# Adapting in the Dark: Efficient and Stable Test-Time Adaptation for Black-Box Models

## Abstract

Test-Time Adaptation (TTA) for black-box models accessible only via APIs remains a largely unexplored challenge. Existing approaches such as post-hoc output refinement offer limited adaptive capacity, while Zeroth-Order Optimization (ZOO) enables input-space adaptation but faces high query costs and optimization challenges in the unsupervised TTA setting. We introduce **BETA** (Black-box Efficient Test-time Adaptation), a framework that addresses these limitations by employing a lightweight, local white-box steering model to create a tractable gradient pathway. Through a prediction harmonization technique combined with consistency regularization and prompt learning-oriented filtering, BETA enables stable adaptation with no additional API calls and negligible latency beyond standard inference. On ImageNet-C, BETA achieves a +7.1% accuracy gain on ViT-B/16 and +3.4% on CLIP, surpassing strong white-box and gray-box methods including TENT and TPT. On a commercial API, BETA achieves comparable performance to ZOO at $250\times$ lower cost while maintaining real-time inference speed, establishing it as a practical and efficient solution for real-world black-box TTA. *Code will be released.*

## 1. Introduction

Modern deep learning models often suffer performance degradation when deployed in the wild due to distribution shifts between training and test data (Recht et al., 2019; Koh et al., 2021). Test-Time Adaptation (TTA) (Sun et al., 2020; Wang et al., 2021; Niu et al., 2023; Wang et al., 2022; Manli et al., 2022) addresses this by adapting a pre-trained model on-the-fly using unlabeled target data. However, feasible adaptation strategies depend critically on the level of model access. While white-box methods assume full access to parameters and gradients (Wang et al., 2021; Niu et al., 2023), many state-of-the-art (SOTA) models are now deployed as opaque APIs (Hurst et al., 2024; Team et al., 2023), where users can only submit inputs and receive output predictions.

We study TTA in this *strict black-box setting*. Concretely, from the user's perspective, one can only send a raw image to the API and receive output prediction probabilities; the model's architecture, parameters, and pre-training data remain entirely unknown. Beyond the adaptation challenge itself, each API call incurs monetary cost and network latency, making query efficiency a first-class concern. An effective black-box TTA method must therefore be not only accurate but also economical, ideally requiring only a single API call per test sample. This setting is fundamentally more challenging than both white-box TTA and supervised black-box adaptation. White-box methods (Wang et al., 2021; Niu et al., 2023; Wang et al., 2022) rely on backpropagation through model parameters, which is impossible without internal access. Supervised approaches such as BlackVIP (Oh et al., 2023) use labeled support sets to guide prompt learning, but TTA must operate on unlabeled test streams without ground-truth. The combination of no gradient access and no labels makes black-box TTA particularly difficult.

Recent backpropagation-free methods (Niu et al., 2024; Karmanov et al., 2024; Lee et al., 2025; Zhou et al., 2025) improve efficiency but still require access to internal tokens or features, placing them in a "gray-box" category (Table 1). Truly black-box methods are scarce and can be categorized by whether they modify the output or the input. Output modification methods like LAME (Boudiaf et al., 2022) refine predictions post-hoc but offer limited adaptive capacity. Input modification methods are more powerful but face practical constraints. Augmentation-based approaches (Farina et al., 2024) aggregate predictions over multiple augmented views, but this linearly increases API costs. Purification-based methods (Gao et al., 2023) use diffusion models to denoise inputs, but they require training on the source domain data and introduce substantial inference latency due to iterative denoising. Visual prompting via Zeroth-Order Optimization (ZOO) (Liu et al., 2018; Spall, 1992; 1997; Hansen & Ostermeier, 2001; Hansen et al., 2003) can learn

[1]Anonymous Institution, Anonymous City, Anonymous Region, Anonymous Country. Correspondence to: Anonymous Author <anon.email@domain.com>.

Preliminary work. Under review by the International Conference on Machine Learning (ICML). Do not distribute.

*Table 1.* Comparison of TTA methods across key capabilities. We evaluate each method's requirements for accessing model parameters, internal tokens, intermediate features, and gradients, alongside its visual encoder architectural flexibility, support for different model types (Vision models (VMs)/Vision-Language models (VLMs)), query efficiency (One API call per test sample), and inference latency.

| Access | Method | w/o Params. | w/o Tokens | w/o Feats. | w/o Grad. | Arch-Agnostic | VMs | VLMs | 1 API/Sample | Low Latency |
|---|---|---|---|---|---|---|---|---|---|---|
| ▭ | TENT (Wang et al., 2021) | ✗ | ✗ | ✗ | ✗ | ✓ | ✓ | ✓ | ✓ | ✓ |
|  | TPT (Manli et al., 2022) | ✗ | ✗ | ✗ | ✗ | ✓ | ✗ | ✓ | ✓ | ✓ |
| ▬ | T3A (Iwasawa & Matsuo, 2021) | ✗ | ✓ | ✗ | ✓ | ✓ | ✓ | ✓ | ✗ | ✓ |
|  | FOA (Niu et al., 2024) | ✓ | ✗ | ✗ | ✓ | ViT-only | ✓ | ✓ | ✗ | ✗ |
|  | B$^2$TPT (Meng et al., 2025) | ✓ | ✗ | ✓ | ✓ | ViT-only | ✗ | ✓ | ✗ | ✗ |
|  | BCA (Zhou et al., 2025) | ✓ | ✓ | ✗ | ✗ | ✓ | ✓ | ✓ | ✓ | ✓ |
| ▬ | LAME (Boudiaf et al., 2022) | ✓ | ✓ | ✓ | ✓ | ✓ | ✓ | ✓ | ✗ | ✓ |
|  | Augmentation (Farina et al., 2024) | ✓ | ✓ | ✓ | ✓ | ✓ | ✓ | ✓ | ✗ | ✗ |
|  | Purification (Gao et al., 2023) | ✓ | ✓ | ✓ | ✓ | ✓ | ✓ | ✓ | ✗ | ✗ |
|  | ZOO | ✓ | ✓ | ✓ | ✓ | ✓ | ✓ | ✓ | ✗ | ✗ |
|  | **BETA (Ours)** | ✓ | ✓ | ✓ | ✓ | ✓ | ✓ | ✓ | ✓ | ✓ |

input perturbations without gradients, but suffers from prohibitive query costs and catastrophic instability when guided by noisy unsupervised signals (Zhang et al., 2024b; Wang et al., 2024b). For instance, accuracy on the Contrast corruption collapses from 32.6% to 4.1% with ZOO (Table 2).

We propose Black-box Efficient Test-time Adaptation (**BETA**), a framework that achieves stable and efficient adaptation using a local, lightweight steering model. This steering model is initialized from public checkpoints and operates entirely on the client side, requiring no access to the target model's internals or pre-training data. Since naive gradient transfer between different architectures is ineffective (Fig. 2), BETA instead employs *prediction harmonization* to fuse outputs from both models, creating a shared objective optimized through the steering model's gradient pathway. To address the instability of learning prompts from random initialization (Fig. 3a), we introduce *consistency regularization* and *prompt learning-oriented filtering*, which together ensure robust adaptation without any ground-truth labels.

BETA achieves strong results across diverse settings. On ImageNet-C with ViT-B/16, it achieves 62.6% accuracy (+7.1% gain), outperforming white-box methods like TENT (Wang et al., 2021) and CoTTA (Wang et al., 2022) with only a single API call per sample. On CLIP, BETA reaches 63.4% accuracy, surpassing specialized VLM methods including TPT (Manli et al., 2022), DynaPrompt (Xiao et al., 2025), and TCA (Wang et al., 2024a). On the real-world commercial Clarifai API, BETA achieves a **+5.2%** gain for just $0.4, while ZOO requires over $100 for comparable performance, demonstrating a **250×** cost advantage.

**Contributions.** (1) We provide the first systematic study of TTA in the strict black-box setting, revealing the limitations of existing approaches including post-hoc refinement, augmentation, purification, and ZOO-based prompting. (2) We introduce BETA, which bypasses expensive queries by using a steering model with prediction harmonization, stabilized by consistency regularization and prompt-oriented filtering for fully unsupervised adaptation. (3) We establish new SOTA results for black-box TTA. BETA surpasses strong

white-box methods with no additional API calls and negligible latency beyond standard inference, making it practical for real-world deployment.

## 2. Related Works

**Test-time Adaptation (TTA).** TTA adapts pre-trained models on-the-fly using unlabeled target data to handle distribution shifts (Sun et al., 2020; Niu et al., 2023; 2022; Wang et al., 2022; Zhang et al., 2025a;b; Manli et al., 2022). Most methods assume *white-box* access, updating model parameters via entropy minimization or consistency objectives (Wang et al., 2021; Niu et al.; 2023). Recent backpropagation-free methods improve efficiency but still require access to internal tokens or features, placing them in a *gray-box* category (Niu et al., 2024; Zhou et al., 2025; Wang et al., 2024a; Lee et al., 2025). Truly *black-box* TTA, where only inputs and outputs are accessible, remains underexplored. Methods operating solely on inputs or outputs are potentially applicable, but output modification offers limited adaptive capacity (Boudiaf et al., 2022), while input modification faces challenges in query efficiency and optimization stability (Gao et al., 2023; Farina et al., 2024).

**Black-box Model Adaptation.** Adapting black-box models has been explored in vision and language domains (Sun et al., 2024; Tsai et al., 2020; Oh et al., 2023; Liu et al., 2024; Sun et al., 2022), but typically for offline transfer learning with labeled support sets. This supervised setting differs fundamentally from unsupervised, online TTA. A prominent approach uses ZOO to learn input prompts for downstream tasks (Oh et al., 2023; Tsai et al., 2020; Liu et al., 2020), but these methods suffer from high query costs and optimization instability, particularly in the unsupervised TTA setting (Wang et al., 2024b; Oh et al., 2023). Other VLM adaptation methods require access to intermediate representations like text embeddings (Ouali et al., 2023; Wang et al., 2024b), violating the strict black-box assumption.

Among input modification strategies, test-time augmentation aggregates predictions over multiple views but linearly increases API costs (e.g., 64×), and some variants

*Figure 1.* Comparison of black-box test-time adaptation strategies. **(a)** Output Refinement (LAME) is limited to post-processing predictions, while **(b)** ZOO-based Input Prompt Learning requires multiple expensive API calls for prompt optimization. In contrast, **(c)** BETA achieves efficient single-query adaptation by leveraging a lightweight steering model with prediction harmonization to create a tractable gradient pathway, stabilized through data filtering and regularization.

require labeled pre-training data (Shanmugam et al., 2021) or logit access for temperature calibration (Farina et al., 2024). Diffusion-based purification (Gao et al., 2023; Nie et al., 2022) reconstructs inputs via generative models, but requires training on source data and introduces substantial latency due to iterative denoising, making it unsuitable for real-time adaptation. In contrast, BETA is the first method to address *unsupervised, online* TTA in the strict black-box setting with both query efficiency and low latency.

## 3. Method

### 3.1. Problem Formulation and Motivation

TTA aims to adapt a model $f$, pre-trained on a source domain, to an unlabeled target domain $\mathcal{D}_T = \{x_j^T\}_{j=1}^{|\mathcal{D}_T|}$ encountered during inference. In the online setting, target data arrives as a stream of batches $\{B_t\}_{t=1}^T$, and the model is updated on-the-fly without ground-truth labels. The feasible adaptation strategies depend on the level of model access, which falls into three categories (Table 1):

- **White-Box Access ( ▢ ):** The full model architecture and all its weights are accessible. This allows for the computation of gradients via backpropagation.

- **Gray-Box Access ( ▨ ):** Intermediate representations, e.g., internal tokens or features, are accessible, while the full computational graph and partial model parameters remain hidden.

- **Black-Box Access ( ▉ ):** The model is an opaque API. Users can only send raw input $x$ and receive output probability vector $p(y|x) = f(x)$. The model's architecture, parameters, pre-training data, training algorithms, and intermediate states are entirely unknown.

**Existing Approaches and Their Limitations.** Black-box TTA methods operate on either the output or input space, each with distinct limitations. *Output refinement* methods like LAME (Boudiaf et al., 2022) are efficient since they

require no additional queries, but their adaptive capacity is inherently limited by working only on final predictions.

*Input modification* methods offer greater adaptive potential but face practical constraints. Augmentation-based approaches (Shanmugam et al., 2021; Farina et al., 2024) aggregate predictions over multiple views, but this linearly increases API costs (e.g., $N$ views require $N$ calls). Purification-based methods (Gao et al., 2023; Nie et al., 2022) use diffusion models to project inputs onto the source manifold, but iterative denoising introduces high latency unsuitable for real-time adaptation. ZOO-based prompting (Niu et al., 2024) learns input perturbations without gradients, but suffers from high query complexity and optimization instability in the unsupervised TTA setting.

### 3.2. BETA: Black-box Efficient Test-time Adaptation

The limitations of existing approaches motivate **BETA**, which seeks to combine the adaptive capacity of input prompting with the query efficiency of output-based methods. To address the inaccessibility of the target model's gradients while avoiding the high cost of ZOO, BETA operates with two models: a *target model* $f_B$, the powerful but inaccessible black-box API (e.g., Clarifai API) from which we can only obtain predictions $p_B(x) = f_B(y|x)$, and a *steering model* $f_S$, a lightweight local model (e.g., ViT-Small) with full access to parameters and gradients.

To adapt the black-box model without altering its weights, we learn an additive visual prompt $\delta \in \mathbb{R}^{H \times W \times C}$. This prompt is added to the input image $x$ to produce a prompted version $x' = x + \delta$. The goal is to optimize $\delta$ using gradients derived locally from $f_S$ to improve the predictions.

**The Challenge of Black-Box Prompt Optimization.** A straightforward approach to optimize $\delta$ in the black-box setting is to employ ZOO to minimize the Shannon entropy of the model's predictions (Wang et al., 2021): $\mathcal{H}(p_B(x')) = -\sum_{k=1}^K p_B^k(x') \log p_B^k(x')$, where $p_B^k(x')$ is the predicted

probability for class $k$. However, our investigation reveals two critical drawbacks. First, ZOO incurs prohibitively high query complexity and latency; for example, a standard CMA-ES setup requires 28 API queries per test sample (Niu et al., 2024). Second, ZOO exhibits fundamental instability when guided by unsupervised signals like entropy, often learning degenerate solutions that corrupt the input's semantic content to produce high-confidence but incorrect predictions. While FOA (Niu et al., 2024) mitigates this by aligning source and target feature statistics, this requires access to source domain data, which is unavailable in the strict black-box setting where the API's training data (e.g., for CLIP from OpenAI) remains proprietary. This instability leads to catastrophic collapse on challenging domains: on the Contrast corruption, accuracy drops from 32.6% to 4.1%, 26.8%, and 12.7% across three ZOO methods (Table 2).

### 3.3. Prediction Harmonization

**From Naive Transfer to Harmonized Relaxation.** Our approach is motivated by the failure of direct gradient estimation. To formalize the analysis, we use $\nabla \mathcal{H}(p; \cdot)$ to denote the gradient of the entropy of prediction $p$, computed by backpropagating through the model indicated by the second argument. Our ultimate goal is to minimize the entropy of the black-box model, which requires following the *target gradient* $g_{\text{Black}} = \nabla \mathcal{H}(p_B; f_B)$. However, since $f_B$ is inaccessible, $g_{\text{Black}}$ is intractable. Existing alternatives fail to provide reliable substitutes: ZOO suffers from costs and instability, while naively transferring the *local gradient* $g_{\text{Local}} = \nabla \mathcal{H}(p_S; f_S)$ from the steering model is ineffective, as our analysis shows the gradient similarity between different architectures is consistently near zero ($\approx 0.0006$).

To overcome this, we relax the problem to finding a prompt that improves both models simultaneously. We define a *harmonized prediction* $p_H$ that fuses the outputs of the steering and target models with a weighting parameter $\alpha \in [0, 1]$:

$$p_H(x') = \alpha \cdot p_S(x') + (1 - \alpha) \cdot p_B(x'). \quad (1)$$

The ideal update direction $g_{\text{Ideal}} = \nabla_\delta \mathcal{H}(p_H; f_S, f_B)$ requires backpropagating through both models. Since $f_B$ is inaccessible, $g_{\text{Ideal}}$ remains intractable. We therefore employ an asymmetric optimization strategy: we compute the gradient of the same harmonized objective but restrict gradient flow exclusively to the steering model's pathway. This yields our tractable proxy $g_{\text{BETA}} = \nabla_\delta \mathcal{H}(p_H; f_S)$, which targets the joint harmonized distribution without requiring internal access to the black-box model.

**Empirical Justification.** To validate $g_{\text{BETA}}$ as a proxy for $g_{\text{Ideal}}$, we conduct a gradient analysis across four validation corruption domains using ViT-B/16 and ViT-L/16 as black-box target models and ViT-S/16 as the local steering model. For this analysis only, we temporarily assume white-

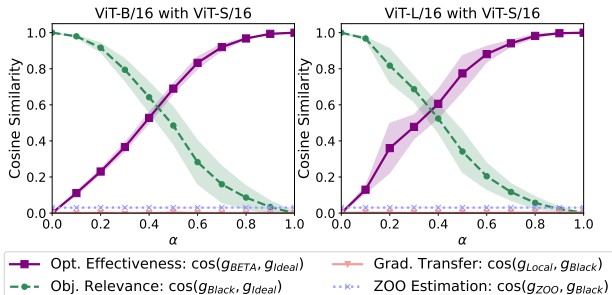

*Figure 2.* We analyze the trade-off between Objective Relevance (alignment with the true target gradient) and Optimization Effectiveness (alignment with the practical steering gradient) as a function of $\alpha$. The intersection of these opposing curves identifies the optimal range (e.g., $\alpha \in [0.3, 0.5]$) where the objective is simultaneously relevant to the target and tractable for optimization. The curves are plotted based on the validation sets of ImageNet-C.

box access to the target models to compute the otherwise inaccessible vectors $g_{\text{Black}}$ and $g_{\text{Ideal}}$. As shown in Fig. 2, simpler strategies fail: the cosine similarity between $g_{\text{Local}}$ and $g_{\text{Black}}$ is consistently near zero for both target models, confirming that naive gradient transfer across different architectures is ineffective. ZOO gradient estimates are equally noisy in the one-step setting despite their high query cost.

BETA's effectiveness stems from how $\alpha$ navigates a trade-off between two competing factors. *Objective Relevance* measures how well our tractable objective aligns with the true goal: Relevance($\alpha$) = $\cos(g_{\text{Ideal}}, g_{\text{Black}})$. *Optimization Effectiveness* measures how well our practical proxy can optimize this objective: Effectiveness($\alpha$) = $\cos(g_{\text{BETA}}, g_{\text{Ideal}})$. These factors oppose each other: low $\alpha$ yields high Relevance but negligible Effectiveness since gradients cannot flow through $f_B$, while high $\alpha$ yields perfect Effectiveness for an irrelevant objective. BETA identifies an optimal range (e.g., $\alpha \in [0.3, 0.5]$) where both are balanced. This confirms that BETA succeeds not by approximating the target gradient directly, but by constructing a shared optimization problem where $g_{\text{BETA}}$ aligns with $g_{\text{Ideal}}$.

### 3.4. Stabilization and Joint Optimization

**Instability of Unconstrained Optimization.** While the harmonized objective provides a tractable gradient pathway, our investigation reveals that this process is inherently unstable when applied in isolation. We evaluated a baseline using only the harmonized objective (Eqn. 1) on ImageNet-C Contrast with ViT-B/16 as the target model across five independent runs. As shown in Fig. 3a, naively optimizing the randomly initialized prompt leads to either gradual performance decay or catastrophic collapse. This instability arises because noisy unsupervised signals can cause the optimization to learn degenerate solutions that corrupt the input's semantic features. To ensure robust adaptation, BETA incorporates two stabilization mechanisms.

*Table 2.* Classification accuracy (%) on ImageNet-C (severity 5) using **ViT-B/16** (87M) as the black-box model. BETA achieves the highest performance among black-box methods and outperforms several strong white-box approaches. *White-box and gray-box methods are shown for reference.* Within black-box methods, **bold** indicates best and underline indicates second best.

| Access | Method | Noise | | | Blur | | | | Weather | | | | Digital | | | | Avg. | Gain |
|---|---|---|---|---|---|---|---|---|---|---|---|---|---|---|---|---|---|---|
| | | Gauss. | Shot | Impul. | Defoc. | Glass | Motion | Zoom | Snow | Frost | Fog | Bright. | Contr. | Elastic | Pixel. | JPEG | | |
| | Source | 56.8 | 56.8 | 57.5 | 46.9 | 35.6 | 53.1 | 44.8 | 62.2 | 62.5 | 65.7 | 77.7 | 32.6 | 46.0 | 67.0 | 67.6 | 55.5 | 0.0 |
| □ | TENT | 60.3 | 61.6 | 61.8 | 59.2 | 56.5 | 63.5 | 59.1 | 54.2 | 64.5 | 2.2 | 79.1 | 67.4 | 61.5 | 72.5 | 70.6 | 59.6 | +4.1 |
| | SAR | 59.1 | 60.5 | 60.6 | 57.1 | 55.6 | 61.5 | 57.4 | 65.8 | 63.4 | 67.4 | 78.7 | 62.6 | 62.2 | 72.0 | 70.2 | 63.6 | +8.1 |
| | CoTTA | 63.3 | 63.9 | 64.5 | 55.0 | 51.0 | 63.5 | 56.1 | 68.8 | 69.2 | 71.2 | 78.3 | 9.6 | 64.3 | 73.4 | 71.2 | 61.6 | +6.1 |
| | ETA | 60.9 | 62.2 | 62.2 | 59.5 | 57.4 | 63.6 | 60.1 | 68.3 | 65.8 | 71.5 | 79.3 | 66.9 | 64.9 | 72.9 | 71.1 | 65.8 | +10.3 |
| ▨ | T3A | 56.4 | 56.9 | 57.3 | 47.9 | 37.8 | 54.3 | 46.9 | 63.6 | 60.8 | 68.5 | 78.1 | 38.3 | 50.0 | 67.6 | 69.1 | 56.9 | +1.4 |
| | FOA* | 57.0 | 58.5 | 57.8 | 51.7 | 35.0 | 37.1 | 27.2 | 20.2 | 11.9 | 72.2 | 76.8 | 0.6 | 39.1 | 66.7 | 67.0 | 44.9 | -10.6 |
| ■ | LAME | 56.5 | 56.5 | 57.2 | 46.4 | 34.7 | 52.7 | 44.2 | 58.4 | 61.5 | 63.1 | 77.4 | 24.7 | 44.6 | 66.6 | 67.2 | 54.1 | -1.4 |
| | ZOO-CMA | 58.2 | 59.6 | 60.3 | 50.8 | 38.6 | 55.2 | 45.7 | 58.5 | 59.6 | 59.7 | 76.7 | 4.1 | 49.8 | 71.2 | 70.0 | 54.5 | -1.0 |
| | ZOO-RGF | 59.6 | 58.7 | 60.4 | 47.7 | 37.8 | 53.5 | 44.6 | 58.2 | 61.7 | 63.4 | 76.7 | 26.8 | 49.4 | 70.7 | 70.2 | 56.0 | +0.5 |
| | ZOO-SPSA-GC† | 59.6 | 58.7 | 60.2 | 47.9 | 38.0 | 53.7 | 44.7 | 58.2 | 61.7 | 63.6 | 76.7 | 12.7 | 49.4 | 70.7 | 70.2 | 55.1 | -0.4 |
| | TT-Aug‡ | 55.4 | 54.2 | 55.2 | 43.7 | 48.6 | 48.9 | 45.5 | 57.8 | 63.1 | 60.0 | 76.9 | 49.6 | 41.7 | 65.7 | 67.8 | 55.6 | +0.1 |
| | DDA§ | **64.7** | **65.0** | **64.6** | 46.3 | 41.3 | 54.4 | 43.7 | 59.1 | 61.3 | 45.0 | 74.9 | 40.6 | **54.4** | **72.2** | 56.9 | 56.9 | +1.4 |
| | **BETA (Ours)** | 60.5 | 60.7 | 61.1 | 54.5 | 52.2 | 59.9 | 56.3 | 63.6 | 64.7 | 66.1 | 78.1 | 53.4 | 62.1 | 73.3 | 72.0 | 62.6 | +7.1 |

*FOA (Niu et al., 2024) uses entropy minimization as the original activation discrepancy requires source statistics unavailable in the black-box setting.

†ZOO-SPSA-GC adapted from (Oh et al., 2023); uses entropy instead of cross-entropy as no ground-truth labels are available.

‡TT-Aug adapted from Shanmugam et al. (2021) which requires labeled source data; we only aggregate predictions over augmented views.

§DDA requires training a diffusion model on source data (Gao et al., 2023); we use the released model as source data is typically unavailable.

**Prompt Learning-oriented Data Filtering.** The first mechanism filters the training signal. Updating the prompt using all incoming data degrades performance because high-entropy samples provide noisy gradients. We therefore update the prompt using only samples with prediction entropy $\mathcal{H}(p_S(x))$ below a threshold $\epsilon$. This filtering is integrated into the harmonization objective via a weight term:

$$\mathcal{L}_{\text{Harmon}}(x') = w_H(x')\mathcal{H}(p_H(x')), \quad (2)$$

where $w_H(x) = \frac{1}{\exp[\mathcal{H}(p_S(x))-\epsilon]} \cdot \mathbb{I}_{\{\mathcal{H}(p_S(x))<\epsilon\}}(x)$ filters out high-entropy samples and assigns confidence-based weights to reliable ones. Unlike methods that filter for pre-trained normalization parameters, we retain all reliable samples including redundant ones for the prompt update, as learning a visual prompt from random initialization is more challenging and benefits from more data.

**Consistency Regularization.** While filtering removes noisy samples, the optimization itself requires regularization to prevent collapse. Since prompts are randomly initialized, an unconstrained entropy objective can be minimized by learning degenerate solutions that destroy the model's representations (Fig 3a). We introduce a consistency regularization that anchors updates to the model's pre-trained knowledge by minimizing the KL-divergence between predictions on clean ($x$) and prompted ($x'$) images:

$$\mathcal{L}_{\text{consist}}(x, x') := D_{\text{KL}}(p_S(x)\|p_S(x')) = \sum_{k=1}^{K} p_S^k(x) \log \frac{p_S^k(x)}{p_S^k(x')}. \quad (3)$$

**Final Objective and Joint Optimization.** BETA operates in a strictly online, one-pass manner, performing a single gradient step per batch $B_t$ to ensure minimal latency. The optimization targets are distinct: the harmonization loss and consistency regularization update the visual prompt $\delta$, while the steering loss updates only the normalization parameters $\theta$ of the steering model. Following Niu et al. (2022), we

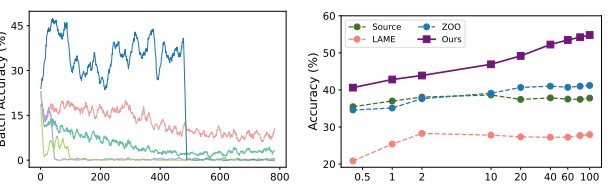

*(a)* Online Batch Acc.      *(b)* Clarifai API Budget ($)

*Figure 3.* (a) Five independent runs using solely Eqn. (1) on ImageNet-C (Contrast, level 5) with ViT-B/16 as the target model, showing either performance collapse or failure to improve. (b) Performance vs. API budget on the real-world Clarifai API.

adapt the normalization layers using only samples that are both reliable and non-redundant, where non-redundancy is determined by comparing a sample's prediction against an exponential moving average of past predictions $\bar{p}_{t-1}$ using a diversity margin $d$. The steering loss is:

$$\mathcal{L}_{\text{Steer}}(x') = w_S(x')\mathcal{H}(p_S(x')), \quad (4)$$

where $w_S(x) = \frac{1}{\exp[\mathcal{H}(p_S(x))-\epsilon]} \cdot \mathbb{I}_{\{\mathcal{H}(p_S(x))<\epsilon\}} \cdot \mathbb{I}_{\{|\cos(p_S(x),\bar{p}_{t-1})|<d\}}$ selects samples that are both reliable (low entropy) and non-redundant (high cosine distance). The final objective combines all components:

$$\mathcal{L}_{\text{BETA}} = \mathbb{E}_{x \in B_t}\left[\mathcal{L}_{\text{Harmon}}(x') + \mathcal{L}_{\text{Steer}}(x') + \lambda\mathcal{L}_{\text{consist}}(x,x')\right]. \quad (5)$$

# 4. Experiments

**Datasets and Models.** We evaluate our method across several challenging benchmarks: ImageNet-C at severity level 5 (Hendrycks & Dietterich, 2019), ImageNet-S (Sketch) (Wang et al., 2019), and ImageNet-R (Rendition) (Hendrycks et al., 2021). In our experiments, we treat powerful, large-scale models as the inaccessible black-box targets: standard Vision Transformers ViT-B/16

*Table 3.* Classification accuracy (%) on ImageNet-C (severity 5) using **ViT-L/16** (304M) as the black-box model. BETA achieves the best performance among black-box methods and outperforms several strong white-box approaches. *White-box and gray-box methods are shown for reference*. Within black-box methods, **bold** indicates best and underline indicates second best.

| Access | Method | Noise | | | Blur | | | | Weather | | | | Digital | | | | Avg. | Gain |
|---|---|---|---|---|---|---|---|---|---|---|---|---|---|---|---|---|---|---|
| | | Gauss. | Shot | Impul. | Defoc. | Glass | Motion | Zoom | Snow | Frost | Fog | Bright. | Contr. | Elastic | Pixel. | JPEG | | |
| | Source | 62.5 | 62.0 | 63.3 | 52.9 | 45.3 | 60.7 | 55.2 | 66.0 | 62.3 | 62.6 | 79.9 | 40.1 | 56.2 | 74.3 | 72.8 | 61.1 | 0.0 |
| ☐ | TENT | 67.2 | 67.3 | 65.4 | 59.2 | 0.9 | 66.7 | 63.8 | 69.7 | 67.0 | 61.9 | 81.0 | 60.3 | 65.4 | 77.3 | 74.1 | 63.1 | +2.0 |
| | SAR | 65.6 | 66.7 | 66.9 | 58.6 | 57.8 | 60.5 | 61.0 | 69.3 | 67.0 | 68.1 | 81.0 | 60.2 | 61.8 | 76.8 | 74.3 | 66.4 | +5.3 |
| | CoTTA | 68.3 | 69.7 | 69.9 | 57.1 | 54.2 | 53.5 | 63.2 | 72.5 | 70.4 | 26.2 | 80.9 | 53.5 | 65.6 | 77.1 | 74.9 | 63.8 | +2.7 |
| | ETA | 67.4 | 58.3 | 67.9 | 63.4 | 61.3 | 67.7 | 62.9 | 70.7 | 68.4 | 66.3 | 81.3 | 54.0 | 66.0 | 77.7 | 74.1 | 67.2 | +6.1 |
| ▨ | T3A | 62.6 | 62.2 | 63.5 | 54.0 | 46.1 | 61.3 | 56.4 | 66.6 | 63.2 | 57.3 | 79.9 | 39.1 | 58.9 | 74.6 | 73.3 | 61.3 | +0.2 |
| | FOA* | 48.1 | 56.1 | 59.1 | 50.2 | 50.6 | 59.6 | 42.4 | 57.5 | 58.8 | 56.1 | 72.2 | 29.1 | 59.5 | 72.0 | 70.4 | 56.1 | -5.0 |
| ■ | LAME | 62.2 | 61.6 | 63.0 | 52.4 | 44.9 | 60.3 | 54.8 | 65.5 | 61.7 | 61.7 | 79.8 | 39.9 | 55.4 | 74.1 | 72.4 | 60.6 | -0.5 |
| | ZOO-CMA | 61.7 | 62.5 | 63.1 | 57.1 | 50.4 | 61.6 | 55.4 | 63.9 | 62.5 | 59.5 | 78.4 | 22.5 | 56.5 | 75.8 | 74.2 | 60.3 | -0.8 |
| | ZOO-RGF | 61.3 | 62.9 | 62.2 | 56.9 | 50.9 | 59.5 | 52.5 | 59.0 | 58.9 | 56.9 | 75.7 | 31.2 | 57.1 | 74.7 | 72.4 | 59.5 | -1.6 |
| | ZOO-SPSA-GC† | 62.8 | 63.5 | 63.4 | 57.0 | 52.2 | 59.8 | 55.9 | 59.0 | 59.7 | 61.7 | 75.5 | 43.0 | 59.9 | 75.1 | 72.4 | 61.4 | +0.3 |
| | TT-Aug‡ | 62.9 | 63.1 | 63.3 | 54.1 | 48.4 | 60.8 | 56.0 | 60.8 | 63.8 | 62.8 | 79.2 | 49.5 | 57.4 | 74.2 | 72.9 | 61.9 | +0.8 |
| | DDA§ | **68.0** | **68.3** | **68.0** | 52.8 | 49.8 | 59.3 | 53.8 | 64.3 | 63.4 | 55.8 | 78.0 | 46.9 | 61.1 | **76.4** | 73.1 | 62.6 | +1.5 |
| | **BETA (Ours)** | 63.1 | 64.0 | 63.5 | **59.7** | **55.1** | **63.6** | **59.4** | **66.1** | **65.0** | **66.2** | **80.0** | **55.1** | **65.0** | 76.2 | **74.5** | **65.1** | +4.0 |

(87M parameters) and ViT-L/16 (304M), and the Vision-Language Model CLIP with a ViT-B/16 backbone (CLIP-B/16, 150M) (Dosovitskiy et al., 2021; Radford et al., 2021). Adaptation is guided by a much smaller, fully accessible ViT-S/16 (22M) steering model. To validate BETA in a practical, real-world scenario, we also test it using a commercial Clarifai[1] API, which charges $0.0032 per request.

**Compared Methods.** We conduct our comparison in the *source-free Fully TTA* setting, benchmarking against methods with varying levels of model access. For White-box methods, we include those applicable to both VMs and VLMs (Tent (Wang et al., 2021), T3A (Iwasawa & Matsuo, 2021), SAR (Niu et al., 2023), and CoTTA (Wang et al., 2022)), along with specialized approaches for VLMs (TPT (Manli et al., 2022), DynaPrompt (Xiao et al., 2025), and DPE (Zhang et al., 2024a)). For Gray-box methods, we compare against FOA (for both VMs and VLMs) (Niu et al., 2024) and others specific to VLMs (TDA (Karmanov et al., 2024), B²TPT (Meng et al., 2025), TCA (Wang et al., 2024a), BCA (Zhou et al., 2025), RA-TTA (Lee et al., 2025)). Our primary comparison is against truly *Black-box* methods: the post-hoc refinement method LAME (Boudiaf et al., 2022), three ZOO baselines (CMA-ES, RGF, and SPSA-GC) that we implemented to learn a visual prompt, augmentation-based TT-Aug (Shanmugam et al., 2021) and VLM-specific ZERO (Farina et al., 2024), and purification-based DDA (Gao et al., 2023).

**Implementation Details.** For BETA, we set the weighting parameter $\alpha$ to 0.4. The shared visual prompt $\delta$ is trained with the AdamW optimizer using a learning rate of 0.01. We update only the normalization layers of the local steering model using SGD with a learning rate of $2 \times 10^{-5}$. The weight for the KL consistency regularization $\lambda$ is set to 50, and we set the entropy threshold $\epsilon = 0.9 \times \ln(1000)$ for sample filtering. The visual prompt is structured as a

---

[1] https://www.clarifai.com/

---

padded frame with a width of 16 pixels, amounting to 39,936 learnable parameters, and is initialized from a Gaussian distribution. More details are in Appendix A.

### 4.1. Experimental Results

**Results on ImageNet-C with Vision Models.** We evaluate BETA against white-box, gray-box, and black-box methods on ImageNet-C using ViT-B/16 as the black-box model (Table 2). Among black-box baselines, output refinement methods like LAME fail to improve upon the source model. ZOO-based approaches are inconsistent, often collapsing on challenging domains such as Contrast, and require significantly more API calls. Augmentation-based methods (TT-Aug) incur high query costs with marginal gains, while purification-based methods (DDA) introduce substantial latency and memory overhead due to iterative diffusion. In contrast, BETA consistently improves across all domains with a single API call, surpassing all black-box baselines and outperforming white-box methods like TENT and CoTTA despite operating under much stricter access constraints.

This trend continues with the more powerful ViT-L/16 (Table 3), where BETA again achieves the best black-box performance while ZOO-based methods degrade. Notably, the steering model (ViT-S/16) is substantially weaker than the target model, and even when fully adapted in a white-box setting, its performance remains well below the target's baseline (Appendix Table 10). Yet BETA successfully leverages this weaker model to improve the powerful black-box model, demonstrating that our method discovers and transfers beneficial adaptation signals rather than simply relying on the steering model's predictions.

**Results on ImageNet-S and ImageNet-R.** We further evaluate BETA's generalization on ImageNet-S and ImageNet-R (Table 4). On both datasets with ViT-B/16, BETA significantly improves upon the source model, surpassing black-box baselines and outperforming white-box methods like

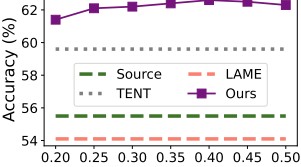

*(a)* Fusion Weight $\alpha$

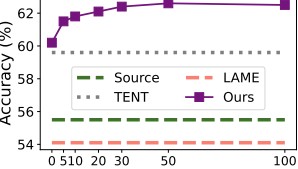

*(b)* KL Reg. Weight $\lambda$

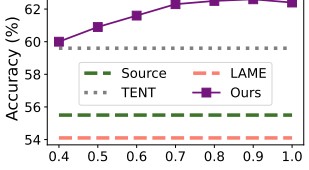

*(c)* Entropy Margin $\epsilon$

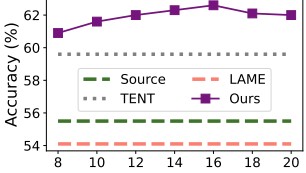

*(d)* Prompt Size

*Figure 4.* Sensitivity analysis of BETA's hyperparameters, showing stable performance across fusion weight $\alpha$ in Eq. 1, regularization weight $\lambda$ in Eq. 5, entropy margin $\epsilon$ in Eq. 2, and prompt size.

*Table 4.* Accuracy (%) on ImageNet-S/R. "-": not applicable.

| Access | Method | ViT-B/16 | | | CLIP (ViT-B/16) | | |
|---|---|---|---|---|---|---|---|
| | | Sketch | Rendition | Avg. | Sketch | Rendition | Avg. |
| | Source | 44.9 | 59.5 | 52.2 | 46.1 | 74.0 | 60.0 |
| ▢ | TENT | 49.1 | 63.9 | 56.5 | 49.5 | 75.3 | 62.4 |
| | SAR | 48.7 | 63.3 | 56.0 | 49.2 | 76.1 | 62.7 |
| | CoTTA | 50.0 | 63.5 | 56.8 | 50.4 | 75.6 | 63.0 |
| | TPT | – | – | – | 48.0 | 77.1 | 62.5 |
| | DynaPrompt | – | – | – | 48.2 | 78.2 | 63.2 |
| | DPE | – | – | – | 52.3 | 80.4 | 66.3 |
| ▨ | T3A | 48.5 | 58.0 | 53.3 | 49.1 | 75.6 | 62.4 |
| | FOA* | 44.7 | 59.2 | 52.0 | 45.8 | 73.2 | 59.5 |
| | TDA | – | – | – | 50.5 | 80.2 | 65.4 |
| | B²TPT | – | – | – | 49.5 | 78.6 | 64.1 |
| | RA-TTA | – | – | – | 50.8 | 79.7 | 65.3 |
| | TCA | – | – | – | 49.0 | 77.1 | 63.0 |
| | BCA | – | – | – | 50.9 | 80.7 | 65.8 |
| ▇ | LAME | 44.4 | 59.0 | 51.7 | 45.4 | 72.8 | 59.1 |
| | ZOO-CMA | 44.7 | 58.8 | 51.8 | 45.6 | 72.5 | 59.1 |
| | ZOO-RGF | 44.4 | 58.1 | 51.3 | 45.3 | 72.1 | 58.7 |
| | ZOO-SPSA-GC† | 45.1 | 59.3 | 52.2 | 46.0 | 72.8 | 59.4 |
| | ZERO¶ | - | - | - | 48.4 | 77.2 | 62.8 |
| | **Ours** | **49.3** | **63.3** | **56.3** | **50.9** | 76.0 | **63.4** |

¶ZERO originally requires logits for re-scaling for CLIP (Farina et al., 2024); we adapt it to use output prediction probabilities.

T3A and SAR. We then extend our evaluation to VLMs, applying BETA to CLIP with a ViT-B/16 backbone. To our knowledge, this is the first work to explore TTA for VLMs in the strict black-box setting. BETA is the only black-box method that effectively improves the pre-trained CLIP model, and remarkably surpasses specialized white-box VLM methods, including TPT and DynaPrompt, as well as gray-box methods such as TCA. This consistent success across datasets and model types demonstrates that BETA is a general framework for black-box adaptation. More results on VLMs are provided in Appendix B.1.

**Results on a Real-world API.** To validate BETA's practicality, we evaluate it on a commercial Clarifai API, comparing performance against API cost on the challenging ImageNet-C Contrast domain (Fig. 3b). The results demonstrate BETA's superior cost-effectiveness. With a budget of just $0.4, sufficient to process approximately 120 test samples, BETA improves upon the source model by +5.2%. In contrast, achieving comparable performance with a query-intensive ZOO method requires over $100, representing a *250× cost advantage* for our approach. Moreover, when given the same $100 budget, BETA's advantage becomes even more pronounced, delivering a substantial +17.1% gain. These results highlight BETA's significant real-world utility, establishing it as a practical and cost-effective solution for

*Table 5.* Efficiency analysis on ImageNet-C (ViT-B/16). BETA matches standard inference in both API cost and latency while achieving the best accuracy with minimal local overhead.

| Method | #API /Img | Local Compute | Mem (MB) | Time (ms) | Acc (%) | Gain (%) |
|---|---|---|---|---|---|---|
| Source | 1 | ✗ | - | 45 | 55.5 | - |
| LAME | 1 | ✓ | 2 | 46 | 54.1 | -1.4 |
| ZOO-SPSA-GC† | 16 | ✓ | 52 | 450 | 55.1 | -0.4 |
| TT-Aug‡ | 64 | ✓ | - | 1,800 | 55.6 | +0.1 |
| DDA§ | 2 | ✓ | 23,427 | 12,722 | 56.9 | +1.4 |
| BETA (ViT-Tiny) | 1 | ✓ | 1,292 | 47 | 58.2 | +2.7 |
| **BETA (ViT-Small)** | **1** | ✓ | 2,616 | 48 | **62.6** | **+7.1** |

adapting commercial API-based models.

**Computational Efficiency.** We analyze BETA's efficiency in Table 5, measuring wall-clock latency and resource usage on a single NVIDIA RTX 3090 GPU. All black-box adaptation methods require local computation, whether CPU-based (e.g., augmentation) or GPU-based (e.g., diffusion or steering models); BETA's overhead remains within reasonable bounds for consumer-grade hardware. The key insight is that local computation is negligible compared to API latency, which dominates total inference time. Methods requiring multiple API calls per image thus suffer severe slowdowns, while BETA matches standard inference in both API cost and latency. More results are in Appendix B.10.

### 4.2. Ablation Studies

**Hyperparameter Sensitivity.** We analyze BETA's sensitivity to its key hyperparameters in Fig. 4. Our analysis of the fusion weight $\alpha$ in Eq. 1 shows that the framework's performance is empirically robust, exhibiting stable and high performance across a wide range of values from 0.3 to 0.5 (Fig. 4a). The KL regularization weight $\lambda$ in Eq. 5 is shown to be a critical component; without it ($\lambda = 0$), performance is suboptimal as the prompt can learn degenerate solutions. As shown in Fig. 4b, performance improves significantly with the introduction of regularization and stabilizes across a broad range of $\lambda$ values from 20 to 100. For the entropy margin $\epsilon$ in Eq. 5, our results show that BETA performs robustly with a more lenient margin (tested from $0.4 \cdot \ln(1000)$ to $1.0 \cdot \ln(1000)$). Unlike methods adapting pre-trained parameters, learning a prompt from random initialization requires more data, making a less restrictive filter beneficial (Fig. 4c). Finally, for the prompt size (Fig. 4d),

*Table 6.* Effect of steering model choice. The Source and TENT-adapted accuracy of each local steering model are provided as a reference against the BETA accuracy on the large black-box models.

| Dataset | Black-Box Model | Source | LAME | ZOO | ViT-Tiny (6M) | | | ResNet50 (26M) | | | ViT-Small (22M) | | |
|---|---|---|---|---|---|---|---|---|---|---|---|---|---|
| | | | | | Source | TENT | **BETA** | Source | TENT | **BETA** | Source | TENT | **BETA** |
| ImageNet-C | ViT-B/16 (87M) | 55.5 | 54.1 | 56.0 | 21.4 | 22.0 | 58.2 | 24.2 | 31.4 | 60.8 | 39.5 | 51.9 | 62.6 |
| ImageNet-Sketch | ViT-B/16 (87M) | 44.9 | 44.4 | 45.1 | 20.9 | 21.3 | 45.2 | 27.9 | 29.7 | 47.5 | 32.8 | 35.6 | 49.3 |
| | CLIP-B/16 (150M) | 46.1 | 45.4 | 46.0 | | | 47.0 | | | 48.7 | | | 50.9 |
| **Average** | - | 48.8 | 48.0 | 49.0 | 21.1 | 21.7 | 50.1 | 26.7 | 30.2 | 52.3 | 35.0 | 41.0 | 54.3 |

*Table 7.* Component analysis. **In-Adapt**: input adaptation via visual prompt learning; **Out-Adapt**: output adaptation via Prediction Refinement (**PR**, LAME) or our Prediction Harmonization (**PH**); **KL Reg.**: consistency KL-regularization; **Filt.**: sample filtering.

| Method | In-Adapt | KL Reg. | Filt. | Out-Adapt | Acc. | Gain |
|---|---|---|---|---|---|---|
| Source | - | - | - | - | 55.5 | 0.0 |
| LAME | - | - | - | PR | 54.1 | -1.4 |
| ZOO | ✓ | - | - | - | 56.0 | +0.5 |
| Exp-1 | - | - | - | PH | 54.2 | -1.3 |
| Exp-2 | ✓ | - | - | PH | 51.6 | -3.9 |
| Exp-3 | ✓ | ✓ | - | PH | 59.7 | +4.3 |
| Exp-4 | ✓ | - | ✓ | PH | 60.2 | +4.7 |
| **BETA** | ✓ | ✓ | ✓ | PH | **62.6** | +7.1 |

which corresponds to the frame width, we observe a clear trade-off: smaller prompts may lack the capacity to capture the domain shift, while larger prompts are harder to optimize. The performance peaks around a width of 16 pixels and remains stable across the tested range of 8 to 20.

**Analysis of BETA's Components.** We conduct an ablation study to dissect each component's contribution, with results in Table 7. Our analysis reveals that strategies focusing solely on output adaptation are insufficient: both LAME's Prediction Refinement and our Prediction Harmonization used in isolation (Exp-1) fail to improve upon the source model, demonstrating that effective black-box TTA requires input adaptation. However, naively adding an input prompt (Exp-2) leads to performance collapse, highlighting the inherent instability of learning a randomly initialized prompt without supervision. Our stabilization techniques address this challenge. Introducing either KL regularization (Exp-3) or sample filtering (Exp-4) provides substantial gains of +4.3% to +4.7%, and the full BETA framework integrating both techniques achieves the best performance at 62.6%. This confirms that both stabilization mechanisms are essential for robust prompt learning.

**Effect of Steering Model Choice.** We investigate how the local steering model affects BETA's performance (Table 6). Our analysis confirms that BETA consistently improves upon the source model across steering models of different sizes and architectures. Even with a 6M-parameter ViT-Tiny, our method successfully boosts large black-box models. The framework also demonstrates cross-architecture generalization, as a CNN-based ResNet-50 can effectively steer Transformer-based ViT and CLIP models. Notably, BETA's improvement far exceeds the capabilities of the

steering models themselves. Even when fully adapted via TENT, our strongest steering model (ViT-Small) attains only 41.0% accuracy, which is significantly lower than the 48.8% black-box baseline. Despite this performance gap, BETA leverages these weaker models to steer the target to 54.3% average accuracy. This demonstrates that BETA effectively discovers and transfers beneficial adaptation signals rather than simply relying on the steering model's predictions.

**Robustness Analysis.** We evaluate BETA's robustness under challenging data stream conditions in the Appendix. Under label imbalance (Niu et al., 2023) and continual domain shifts (Wang et al., 2022), BETA maintains 61.8% and 61.5% accuracy respectively, with minimal degradation from the standard i.i.d. setting (62.6%) (Appendix B.7). BETA also exhibits high robustness to batch size variations, remaining effective even with a batch size of 4 (Appendix B.9). Furthermore, under strict real-time streaming constraints (Alfarra et al.), BETA maintains 62.5% accuracy while ZOO drops to 54.3% due to adaptation lag (Appendix B.10). This robustness stems from BETA's design: the black-box model parameters remain frozen, and adaptation occurs solely through the input prompt with consistency regularization to prevent overfitting to local biases.

## 5. Conclusion

In this work, we addressed the critical challenge of adapting powerful models in the strict black-box setting where only API access is available. We introduced **BETA**, a novel framework that enables efficient and stable Test-Time Adaptation by leveraging a lightweight white-box steering model. The core of our method is a prediction harmonization technique that creates a tractable, shared objective, which is made robust through consistency regularization and a prompt-oriented data filtering strategy. Our extensive experiments show that BETA significantly outperforms existing black-box methods, achieves performance competitive with strong white-box approaches on both Vision and Vision-Language models, and demonstrates immense practical value on a commercial API with a $250\times$ cost advantage over ZOO-based techniques. By demonstrating that a smaller, local model can effectively steer a powerful, inaccessible one, our work makes robust black-box TTA a practical reality and opens up new possibilities for adapting models in the dark at test time.

## Impact Statement

This paper presents work whose goal is to advance the field of Machine Learning. There are many potential societal consequences of our work, none of which we feel must be specifically highlighted here.

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

# **Appendix**

This appendix provides supplementary material organized as follows: Appendix A formalizes the black-box setting and describes all baseline methods; Appendix B.1 presents additional quantitative results and comparisons including knowledge distillation (Appendix B.3) and zeroth-order optimization baselines; Appendix B.7–B.10 evaluate robustness under challenging conditions and computational efficiency; finally, Appendix D discusses limitations of our approach.

## A. Extended Related Work & Black-Box Setting Analysis

### A.1. Detailed Analysis of Model Accessibility and Security Constraints

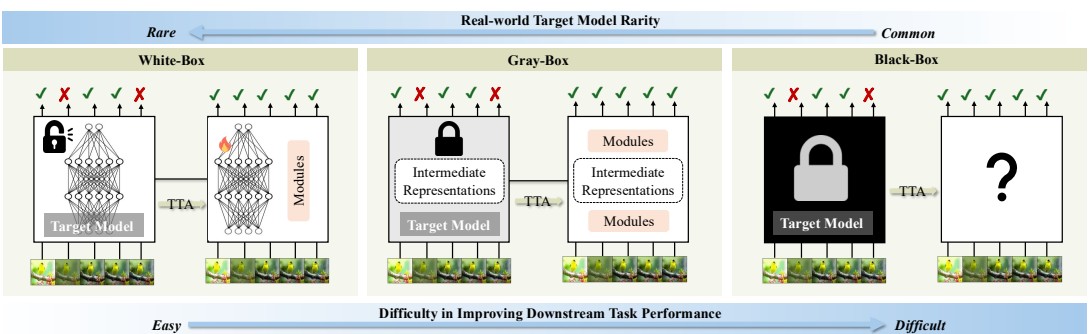

*Figure 5.* The black-box Test-time Adaptation setting studied in this work. From the client's perspective, the goal is to adapt a powerful server-side API model to a target distribution (e.g., corrupted or domain-shifted images) without any internal access. The client can only send raw input images and receive softmax probability vectors in return. Unlike white-box TTA, no gradients, parameters, intermediate features, or architectural details are available, making this a practical yet challenging scenario for real-world API-based deployment.

In this section, we provide a rigorous definition of the black-box setting adopted in this work. While prior literature often conflates different levels of restricted access, we draw sharp distinctions between access to *raw logits*, *softmax probabilities*, and *hard predictions*. This distinction is critical for evaluating the practical applicability of Test-Time Adaptation (TTA) methods on real-world commercial APIs.

**Mathematical Definitions of Output Levels.** Let $f_\theta(x)$ denote the pre-trained model. We distinguish between three specific levels of output granularity:

1. *Raw Logits (z):* The pre-activation output vector $z \in \mathbb{R}^C$, where values are unbounded ($-\infty < z_i < \infty$) and unnormalized.

2. *Softmax Probability Vector (p):* The normalized output distribution obtained via the softmax function $\sigma(\cdot)$, such that $p = \sigma(z) \in [0,1]^C$ with $\sum_i p_i = 1$.

3. *Top-1 Hard Prediction ($\hat{y}$):* A single scalar value representing the class index with the highest confidence, $\hat{y} = \arg\max_i p_i$, often accompanied by a single confidence score.

**Real-World API Protocols.** To determine the most realistic setting for black-box adaptation, we analyze standard commercial Machine Learning APIs (e.g., OpenAI (Hurst et al., 2024), Clarifai, Google Cloud Vision).

- *Why not Raw Logits?* Access to $z$ is frequently restricted as a security measure. Raw logits contain rich information regarding inter-class relationships ("dark knowledge") that significantly facilitates Model Extraction attacks and Knowledge Distillation (Hinton et al., 2015). By hiding $z$, API providers mitigate the risk of model theft.

- *Why Softmax Probabilities?* Most commercial APIs return the probability distribution $p$ rather than a single hard label $\hat{y}$. This is because downstream users typically require confidence estimates to make informed decisions (e.g., thresholding low-confidence predictions).

**Justification for BETA's Setting.** Based on these protocols, we define the strict *Black-Box* setting as one where the *Softmax Probability Vector* $p$ is available, but *Raw Logits* $z$ are hidden. This setting strikes the balance found in real-world

deployments: it provides more information than the restrictive *Label-Only* setting (which only provides $\hat{y}$), enabling unsupervised objectives like entropy minimization ($\mathcal{H}(p) = -\sum p_i \log p_i$). In contrast, we classify methods that require access to raw logits $z$ (e.g., for temperature scaling $z/\tau$ or re-normalization (Farina et al., 2024)) as *Gray-Box*. While these methods do not require gradients, they rely on information often hidden in secure deployment environments.

### A.2. Extended Baselines Description

We compare BETA against a comprehensive suite of baselines with varying levels of model access, including white-box, gray-box, and black-box methods.

**Tent** (Wang et al., 2021) is a **white-box** method for fully test-time adaptation, which adapts a pre-trained model to a new test distribution without requiring any source data. The core idea is to encourage model confidence on the unlabeled test data by minimizing the Shannon entropy of its predictions for each incoming batch. To achieve this efficiently, Tent does not update the entire model; instead, it exclusively adapts the parameters within the model's normalization layers. For each test batch, it first updates the normalization statistics during the forward pass and then optimizes the learnable channel-wise affine transformation parameters via backpropagation on the entropy loss.

**SAR** (Niu et al., 2023) is a **white-box** method designed to stabilize online Test-Time Adaptation in challenging "wild" scenarios, such as with mixed domain shifts or small batch sizes, where standard entropy minimization can fail. The method identifies that model collapse during adaptation is often caused by noisy test samples producing large, disruptive gradients. To mitigate this, SAR employs a two-part strategy: it first filters out unreliable, high-entropy samples to reduce noise. For the remaining data, it then uses a sharpness-aware optimizer to guide the model parameters into a flat region of the loss landscape, enhancing robustness against any remaining noisy updates.

**Continual Test-Time Adaptation (CoTTA)** (Wang et al., 2022) is a **white-box** method designed to adapt models to continually changing target domains, addressing the challenges of error accumulation and catastrophic forgetting. To generate more reliable pseudo-labels, it employs a teacher-student framework where the student model is updated based on the weight-averaged and augmentation-averaged predictions of the teacher. To prevent catastrophic forgetting over long-term adaptation, CoTTA stochastically restores a small fraction of the student model's weights to their original source-trained values during the update process. The method is designed to adapt all parameters of the network.

**Test-Time Template Adjuster (T3A)** (Iwasawa & Matsuo, 2021) is a **gray-box** method for domain generalization that adapts a model's final linear classifier at test time. The method is backpropagation-free and works by first computing class-specific "pseudo-prototype" representations from the features of unlabeled test data. Once these prototypes are established, it classifies each new test sample based on its distance to these dynamically adjusted prototypes. This allows the model to leverage information from the target domain without requiring extensive optimization or altering the core feature extractor.

**Forward-Optimization Adaptation (FOA)** (Niu et al., 2024) is a **gray-box** method designed for test-time adaptation in scenarios where backpropagation is infeasible, such as on quantized models or edge devices. The approach is entirely training-free and avoids modifying model weights by learning an additive input prompt using a derivative-free optimizer (CMA-ES). To guide this optimization, FOA introduces a novel fitness function that combines prediction entropy with a term measuring the statistical discrepancy between the test sample's activations and pre-computed source data activations. The framework also includes a "back-to-source" activation shifting scheme that directly modifies the final layer's features during the forward pass to better align them with the source domain.

**LAME** (Boudiaf et al., 2022) is a **black-box** method for online test-time adaptation that operates without requiring access to model parameters or gradients. Instead of adapting the network's weights, it adapts the model's output probabilities directly for a given batch of test data. The method proposes a Laplacian Adjusted Maximum-likelihood Estimation (LAME) objective, which finds the optimal latent class assignments by maximizing the data likelihood while being regularized by a Laplacian term that encourages label consistency among neighboring samples in the feature space. This objective is optimized efficiently using a concave-convex procedure and does not require backpropagation.

In contrast to the methods above, the following baselines are designed specifically for the adaptation of Vision-Language Models:

**Test-Time Prompt Tuning (TPT)** (Manli et al., 2022) is a **white-box** method that adapts Vision-Language Models like CLIP using only a single unlabeled test sample. For each test image, TPT creates multiple augmented views and optimizes a

learnable text prompt via backpropagation to enforce prediction consistency across them. The optimization is guided by minimizing the entropy of the averaged predictions, and a confidence selection module filters out noisy augmentations that yield low-confidence outputs. TPT performs a one-step update on the prompt for each test sample.

**Dual Prototype Evolving (DPE)** (Zhang et al., 2024a) is a **white-box** method that performs test-time adaptation for VLMs by accumulating task-specific knowledge from both visual and textual modalities. The method maintains and evolves two sets of class prototypes—one textual and one visual—which are updated online as more test samples are processed. For each individual test sample, DPE learns temporary residual parameters to adjust both sets of prototypes. This sample-specific optimization is guided by a dual objective that encourages prediction consistency across augmented views and enforces alignment between the textual and visual prototypes for each class.

**DynaPrompt** (Xiao et al., 2025) is a **white-box** method that improves online test-time prompt tuning by leveraging information from previous test samples while mitigating the problem of prompt collapse. The core of the method is an online prompt buffer containing a set of learnable prompts that evolve over time. For each new test sample, DynaPrompt employs a dynamic selection strategy based on prediction entropy and probability difference to choose a relevant subset of prompts from the buffer for optimization. To adapt to new data, the framework also dynamically appends new prompts to the buffer and removes inactive ones.

**B$^2$TPT** (Meng et al., 2025) is a **gray-box** method that addresses test-time prompt tuning for black-box Vision-Language Models (VLMs) where gradients are inaccessible. To overcome this, it employs a derivative-free algorithm (CMA-ES) to optimize low-dimensional "intrinsic prompts," which are then projected into the full prompt space to make the high-dimensional optimization tractable. For supervision, the framework uses a "Consistent or Confident" (CoC) pseudo-labeling strategy to generate labels from the model's outputs. The method jointly optimizes text and vision prompts using a frozen CLIP ViT-B/16 backbone.

**Training-free Dynamic Adapter (TDA)** (Karmanov et al., 2024) is a **gray-box** method designed for efficient test-time adaptation of Vision-Language Models without requiring backpropagation. The method constructs a lightweight key-value cache during inference, which is progressively updated with incoming test samples. This cache consists of two components: a positive cache that stores image features and their corresponding high-confidence pseudo-labels, and a novel negative cache that stores negative pseudo-labels to improve robustness against label noise. The final prediction is a combination of the original CLIP output and the predictions derived from both the positive and negative caches.

**Retrieval-Augmented TTA (RA-TTA)** (Lee et al., 2025) is a **gray-box** method that adapts Vision-Language Models by incorporating external knowledge from a large image database at test time. Instead of a direct image-to-image search, RA-TTA uses a novel description-based retrieval process to find more relevant external images. For a given test image, it first identifies its most prominent visual features by selecting matching fine-grained text descriptions from a pre-compiled library. These selected text descriptions are then used as queries to retrieve semantically similar images from the database, and the VLM's initial prediction is refined using a relevance score derived from this external knowledge.

**Bayesian Class Adaptation (BCA)** (Zhou et al., 2025) is a **gray-box** method that adapts Vision-Language Models by updating both the class likelihood and prior at test time. It frames the adaptation problem using Bayes' theorem, identifying that existing methods only adapt the likelihood (class embeddings) while overlooking the class prior, which can shift in new domains. BCA employs a dual-update mechanism: it adapts the likelihood by updating the most relevant class embedding with an incoming visual feature via a running average. Concurrently, it adapts the prior by using the model's posterior prediction for the current sample to update the prior distribution of the predicted class, allowing the model to learn the new class frequencies on the fly.

**Token Condensation as Adaptation (TCA)** (Wang et al., 2024a) is a **gray-box** method that provides an efficient, training-free solution for test-time adaptation in Vision-Language Models. The method uniquely repurposes token condensation, a technique originally for improving ViT efficiency, as an adaptation mechanism. It introduces a domain-aware token reservoir that stores reliable class tokens from past test samples to serve as domain anchors. These anchors guide both a cross-head token condensation process, which prunes irrelevant visual tokens, and a logits self-correction mechanism that refines the model's final prediction.

# B. Comprehensive Quantitative Analysis

## B.1. Additional Results on ImageNet Variants and EuroSAT

To provide a comprehensive evaluation, we extend our comparisons to include augmentation-based strategies and recent methods tailored for Vision-Language Models (VLMs). Specifically, we compare BETA against **ZERO** (Farina et al., 2024), a test-time augmentation method that optimizes temperature using input augmentations. We note that while ZERO requires access to raw logits—violating strict black-box API constraints that typically only provide probabilities—we grant it this access for a rigorous upper-bound comparison. We evaluate both the standard ZERO (64 calls/image) and **ZERO_ensemble** (448 calls/image, using 7 text templates). We also include **$B^2$TPT** (Meng et al., 2025), a recent gray-box prompt tuning method for VLMs.

*Table 8.* Performance comparison on ImageNet Variants with CLIP-B/16. BETA outperforms strong augmentation-based and gray-box baselines while requiring only a single API call per image.

| Method | IN-S | IN-R | IN-A | IN-v2 | IN | Avg. | Gain | #API |
|---|---|---|---|---|---|---|---|---|
| Source | 46.1 | 74.0 | 47.9 | 60.9 | 66.7 | 59.1 | – | 1 |
| LAME | 45.4 | 72.8 | 48.1 | 61.6 | 66.7 | 58.9 | -0.2 | 1 |
| ZOO-SPSA-GC | 46.0 | 72.8 | 50.2 | 61.5 | 65.8 | 59.3 | +0.1 | 16 |
| $B^2$TPT (w/ tokens) | 49.5 | 78.6 | 55.3 | 65.4 | 69.6 | 63.7 | +4.6 | 120 |
| ZERO (w/ logits) | 48.4 | 77.2 | 59.6 | 64.2 | 69.3 | 63.7 | +4.6 | 64 |
| ZERO_ens (w/ logits) | 50.6 | 80.8 | 62.8 | 65.2 | 71.2 | 66.1 | +7.0 | 448 |
| **BETA (Ours)** | **50.9** | **76.0** | **62.8** | **65.1** | **77.5** | **66.5** | **+7.4** | **1** |

*Table 9.* Performance on the fine-grained EuroSAT dataset with CLIP-B/16. BETA achieves significant gains (+11.3%) with high efficiency.

| Method | Acc (%) | Gain | #API |
|---|---|---|---|
| Source | 42.0 | – | 1 |
| $B^2$TPT (w/ tokens) | 46.8 | +4.8 | 120 |
| ZERO (w/ logits) | 39.6 | -2.4 | 64 |
| ZERO_ensemble (w/ logits) | 43.8 | +1.8 | 448 |
| **BETA (Ours)** | **53.3** | **+11.3** | **1** |

**Classification of $B^2$TPT as Gray-Box.** We categorize $B^2$TPT as a gray-box method because it operates by modifying inputs in the embedding space. Specifically, it prepends learnable vectors directly to the text and image embeddings ($e_t$ and $e_v$), requiring internal access to the model's intermediate feature representations. This contrasts with the strict black-box setting of commercial APIs, which accept only raw image or text inputs. Furthermore, its underlying optimization (CMA-ES) is query-intensive, requiring approximately 120 API calls per input.

**Results on ImageNet Variants and EuroSAT.** We evaluate these baselines on the full suite of ImageNet variants (ImageNet-S, R, A, v2, and standard ImageNet) and the challenging fine-grained EuroSAT dataset. The results are summarized in Table 8 and Table 9.

BETA consistently outperforms these query-intensive baselines while maintaining strict API efficiency. On the ImageNet variants (Table 8), BETA achieves the highest average accuracy of 66.5%, surpassing the ensemble version of ZERO (66.1%) which requires 448 API calls per image. The efficiency gap is even more pronounced on EuroSAT (Table 9), where BETA achieves a substantial gain of +11.3% over the source model with a single API call, whereas augmentation baselines struggle or yield marginal gains despite their high computational cost. This demonstrates that BETA's effectiveness stems from learned adaptation rather than simple data augmentation, making it a far more practical solution for real-world deployment where API costs and rate limits are critical constraints.

## B.2. White-box TTA performance on Steering Model.

To demonstrate that BETA's improvement is non-trivial and not simply a result of relying on the steering model's outputs, we present the white-box adaptation performance of the ViT-Small steering model in Table 10. There exists a substantial performance gap between the pre-trained steering model (39.5% accuracy on ImageNet-C) and the target black-box models (e.g., ViT-L/16 at 61.1% accuracy). Even when the steering model itself is fully adapted in a white-box setting with a strong method like SAR, its performance is capped at 57.4%. This is still well below the starting accuracy of the black-box model it

is meant to guide. This highlights that BETA successfully leverages this weaker, suboptimal steering model not for its direct predictions, but to discover and transfer beneficial adaptation signals to the far more powerful black-box model without requiring any internal access.

*Table 10.* White-box TTA performance on the ViT-Small steering model (ImageNet-C). Even when fully adapted, the steering model's performance is capped well below that of the unadapted black-box target models (ViT-B/16: 55.5%, ViT-L/16: 61.1%).

|  | Source | TENT | T3A | SAR | CoTTA | LAME |
|---|---|---|---|---|---|---|
| **Avg. Acc (%)** | 39.5 | 51.9 | 40.4 | 57.4 | 46.0 | 38.9 |
| **Gain (%)** | 0.0 | +12.4 | +0.9 | +17.9 | +6.5 | -0.6 |

## B.3. Comparison with Test-Time Knowledge Distillation

A natural question arises as to whether BETA's improvements stem from simply distilling the powerful black-box model's knowledge into the local steering model. To investigate this, and to verify that our framework is not merely performing Test-Time Knowledge Distillation (KD), we implemented a KD baseline following the protocol in (Zhao et al., 2024). Specifically, we employed the black-box ViT-B/16 as the teacher and the local ViT-S/16 as the student, optimizing the student to match the teacher's predictions on the target data.

The results, summarized in Table 11, reveal a fundamental distinction between the two approaches. Standard distillation is inherently limited by the capacity of the student model; the distilled ViT-S/16 achieves only 50.3% accuracy, failing to even match the original performance of the black-box teacher (55.5%). This result is expected, as KD aims to mimic the teacher's existing boundary rather than adapt it to the new domain.

In sharp contrast, BETA achieves 62.6% accuracy, significantly surpassing the original black-box model. This confirms that BETA is not a distillation process where a student mimics a fixed teacher. Instead, BETA utilizes the local model to actively *adapt* the input prompts for the black-box model, allowing the final system to break through the performance ceiling of the original pre-trained weights.

*Table 11.* Comparison between Test-Time Knowledge Distillation and BETA on ImageNet-C. While KD is upper-bounded by the teacher's performance, BETA successfully adapts the black-box model beyond its original baseline.

| Model Role | Architecture | Method | Avg. Acc (%) |
|---|---|---|---|
| Local Steering | ViT-S/16 | Source | 39.5 |
|  |  | TENT | 51.9 |
|  |  | KD (from ViT-B/16) | 50.3 |
| Black-Box Target | ViT-B/16 | Source | 55.5 |
|  |  | **BETA (Ours)** | **62.6** |

## B.4. Zeroth-Order Optimization Baselines

As a direct approach to adapting the visual prompt $\delta$ in a black-box setting, we evaluate several Zeroth-Order Optimization (ZOO) baselines. These derivative-free methods optimize the prompt by minimizing a fitness function, which we define as the Shannon entropy of the black-box model's predictions on the prompted input, $f(\delta) = \mathcal{H}(p_B(x + \delta))$. For a fair comparison, we configure all three ZOO methods to use 16 queries per test sample for their optimization process.

**CMA-ES.** As a representative ZOO method, **Covariance Matrix Adaptation Evolution Strategy (CMA-ES)** is a derivative-free algorithm used to optimize a high-dimensional visual prompt where gradients are inaccessible (Hansen & Ostermeier, 2001; Hansen et al., 2003; Niu et al., 2024; Meng et al., 2025). In each iteration, CMA-ES samples a population of candidate prompts from a multivariate normal distribution and evaluates them using the fitness function. The goal is to find a prompt, $\delta$, that minimizes this entropy, encouraging high-confidence predictions. Based on the performance of the sampled prompts, CMA-ES updates the mean and covariance matrix of the sampling distribution to guide the search towards more promising regions of the solution space.

**RGF Random Gradient-Free (RGF)** is a ZOO method that estimates the gradient of the fitness function by sampling multiple random directions from a standard Gaussian distribution (Liu et al., 2018; Tsai et al., 2020). For a given visual prompt $\delta$, RGF approximates the gradient by averaging the function's response to small perturbations along these random directions, allowing it to descend the loss landscape without direct gradient calculations. The gradient approximation at

iteration $t$ is computed as:

$$g_t(\delta_t) = \frac{1}{q} \sum_{i=1}^{q} \frac{f(\delta_t + \mu u_i) - f(\delta_t)}{\mu} u_i \tag{6}$$

where $u_i$ is a random direction vector drawn from $\mathcal{N}(0, I)$, $\mu$ is a small smoothing parameter, and $q$ is the number of directions sampled.

**SPSA with Gradient Correction (SPSA-GC)** To optimize the visual prompt under black-box constraints, we adopt the Simultaneous Perturbation Stochastic Approximation with Gradient Correction (SPSA-GC) algorithm, as utilized in BlackVIP (Oh et al., 2023). SPSA is a highly efficient ZOO algorithm that estimates the gradient using only two queries per iteration (Spall, 1992). Unlike RGF, which requires sampling multiple directions, SPSA perturbs the parameters in a single random direction and its opposite. The gradient approximation at iteration $t$ for a visual prompt $\delta_t$ is computed as:

$$\hat{g}_t(\delta_t) = \frac{f(\delta_t + \mu \Delta_t) - f(\delta_t - \mu \Delta_t)}{2\mu} \Delta_t \tag{7}$$

where $\Delta_t$ is a random perturbation vector drawn from a Bernoulli distribution, and $\mu$ is a small step size.

While standard SPSA is query-efficient, the stochastic gradient estimate $\hat{g}_t$ can be noisy. To mitigate this, we employ the Gradient Correction mechanism proposed in BlackVIP (Oh et al., 2023). This method integrates Nesterov's Accelerated Gradient (NAG) into the update rule, using a momentum accumulator to rectify the estimated gradient direction. By smoothing the optimization trajectory, SPSA-GC significantly enhances stability compared to vanilla SPSA, making it particularly suitable for the high-dimensional optimization of visual prompts.

### B.5. API efficiency comparison across black-box methods

Table 12 demonstrates BETA's superior efficiency compared to existing black-box TTA methods. While ZOO-based approaches (CMA, RGF, SPSA) require 16 API calls per test sample and achieve modest or negative performance gains ranging from -1.0% to +0.5%, BETA achieves a substantial +7.1% improvement with only a single API call per sample. This represents a $16\times$ reduction in API usage while delivering significantly better adaptation performance. LAME, though equally efficient with one API call, suffers from limited adaptive capacity due to its post-hoc output refinement approach, resulting in a -1.4% performance drop. These results highlight BETA's unique combination of query efficiency and adaptation effectiveness in the black-box setting.

### B.6. Orthogonality of Contribution: Unsupervised Objective vs. ZOO Algorithms

While we adopt the powerful ZOO algorithm like SPSA-GC (Oh et al., 2023) due to its superior efficiency, it is crucial to distinguish the role of the *ZOO algorithm* from the challenges inherent to the *adaptation objective*. The efficacy of SPSA-GC was originally demonstrated in BlackVIP (Oh et al., 2023) within a *supervised* few-shot transfer setting. In that context, the loss landscape is anchored by ground-truth labels via a Cross-Entropy loss, providing a consistent and convex directional signal for the zeroth-order estimator.

In contrast, our strictly **unsupervised online setting** relies on objectives such as entropy minimization. We observe that replacing the supervised loss with an unsupervised one fundamentally alters the optimization landscape, making it prone to trivial solutions. As evidenced in our experimental results, naively applying even a robust ZOO algorithm like SPSA-GC to this unsupervised objective leads to prompt collapse, where the model exploits high-frequency patterns to minimize entropy without preserving semantic integrity. Therefore, we clarify that our primary contribution does not lie in the ZOO algorithm itself. Rather, our contribution is the **unsupervised stabilization framework**: comprising Prediction Harmonization, the Coordinator architecture, and Consistency Regularization. These mechanisms effectively constrain the optimization space, preventing the instability inherent to source-free black-box adaptation and enabling effective Test-Time Adaptation.

### B.7. Robustness to Label Imbalance and Continual Shifts

While our primary evaluation follows the standard episodic adaptation setting, real-world data streams often exhibit temporal correlations or non-stationary distributions. To validate the stability of BETA in dynamic environments, we extend our evaluation on ImageNet-C (using ViT-B/16) to include two challenging scenarios:

- **Label Imbalance** (Niu et al., 2023; Gong et al., 2022): Following the protocol established in SAR (Niu et al., 2023),

*Table 12.* API efficiency comparison across black-box TTA methods. BETA achieves the best accuracy-efficiency trade-off with a single API call per sample.

| Method | #API/sample | Acc (%) | Gain |
|--------|------------|---------|------|
| Source (Inference) | 1 | 55.5 | 0.0 |
| LAME | 1 | 54.1 | -1.4 |
| ZOO-CMA | 16 | 54.5 | -1.0 |
| ZOO-RGF | 16 | 56.0 | +0.5 |
| ZOO-SPSA-GC | 16 | 55.1 | -0.4 |
| TT-Aug | 64 | 55.6 | +0.1 |
| DDA | 2 | 56.9 | +1.4 |
| **BETA (Ours)** | **1** | **62.6** | **+7.1** |

*Table 13.* Robustness analysis on ImageNet-C (ViT-B/16) under Label Imbalance and Continual Domain Shift settings. BETA demonstrates minimal degradation compared to the standard setting, highlighting its stability in dynamic environments.

| Method | Gauss. | Shot | Impul. | Defoc. | Glass | Motion | Zoom | Snow | Frost | Fog | Bright. | Contr. | Elastic | Pixel. | JPEG | Avg. |
|--------|--------|------|--------|--------|-------|--------|------|------|-------|-----|---------|--------|---------|--------|------|------|
| Source | 56.8 | 56.8 | 57.5 | 46.9 | 35.6 | 53.1 | 44.8 | 62.2 | 62.5 | 65.7 | 77.7 | 32.6 | 46.0 | 67.0 | 67.6 | 55.5 |
| BETA (Standard) | 60.5 | 60.7 | 61.1 | 54.5 | 52.2 | 59.9 | 56.3 | 63.6 | 64.7 | 66.1 | 78.1 | 53.4 | 62.1 | 73.3 | 72.0 | 62.6 |
| BETA (**Label Imbalance**) | 59.0 | 59.9 | 59.5 | 53.9 | 51.1 | 59.1 | 55.5 | 62.9 | 64.3 | 65.4 | 77.9 | 52.4 | 61.2 | 73.1 | 72.1 | 61.8 |
| BETA (**Continual Shifts**) | 59.5 | 61.0 | 60.4 | 52.3 | 51.4 | 58.4 | 55.2 | 61.8 | 63.3 | 63.8 | 77.4 | 51.8 | 61.7 | 72.5 | 71.3 | 61.5 |

we evaluate performance on data streams with highly skewed class distributions within each batch, simulating non-i.i.d. test streams.

- **Continual Domain Shifts** (Wang et al., 2022; Niu et al., 2022): Following the Continual Test-Time Adaptation (CoTTA) setting (Wang et al., 2022), the model adapts to the 15 corruption domains of ImageNet-C sequentially without resetting the model state between domains.

The results are summarized in Table 13. BETA exhibits remarkable stability, maintaining high performance even under these challenging conditions. In the label imbalance setting, BETA achieves an average accuracy of 61.8%, and under continual shifts, it maintains 61.5%. This represents minimal degradation compared to the standard i.i.d. setting (62.6%).

**Why is BETA robust?** This robustness is intuitive given our framework's design. Unlike white-box methods that directly update internal model parameters—a process known to risk catastrophic forgetting or overfitting to biased batches—BETA keeps the parameters of the black-box target model frozen. We exclusively learn an additive input prompt. Furthermore, the local steering model is updated with a conservative learning rate and strong consistency regularization, preventing the optimization trajectory from over-fitting to the dynamic changes or local biases in the data stream. This makes BETA naturally resilient to the instability often observed in dynamic test-time adaptation.

### B.8. Analysis on Stabilization Mechanisms

We conduct a component analysis to demonstrate the importance of our two stabilization mechanisms, visualizing the online batch accuracy on the challenging ImageNet-C Contrast domain. The figure shows that the full BETA framework ("Ours") rapidly achieves high accuracy and maintains stable performance across all 800 online batches. In contrast, removing the data filtering component ("w/o Data Filtering") results in significantly lower and gradually decaying performance. More critically, removing the consistency regularization ("w/o KL Reg.") leads to catastrophic collapse, with the model's accuracy plummeting to near zero after approximately 400 batches. This analysis empirically validates that both the consistency regularization and the data filtering are essential for the stable and effective performance of BETA.

### B.9. Robustness to Batch Size

In practical online deployment, the number of samples available for adaptation at any given time step can vary significantly. To assess BETA's sensitivity to this factor, we evaluated its performance on ImageNet-C (ViT-B/16) using batch sizes ranging from 4 to 128. As shown in Table 14, BETA demonstrates high robustness to batch size variations. Even with a very small batch size of 4, where gradient estimates are typically noisy, BETA achieves an average accuracy of 59.3%, significantly outperforming the source model baseline of 55.5%. The performance consistently improves as the batch size increases, saturating at 62.6% for batch sizes of 64 and above. This indicates that while larger batches provide more stable gradients, BETA remains effective even in low-data regimes.

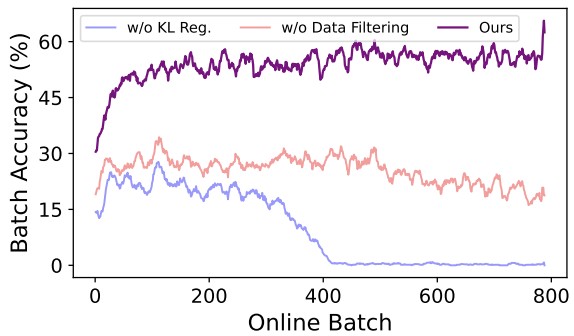

*Figure 6.* Online Batch Accuracy on ImageNet-C Contrast domain.

*Table 14.* Effect of batch size on average accuracy (%) on ImageNet-C. BETA consistently improves upon Source (55.5%) even with extremely small batch sizes.

| Batch Size | Source | 4 | 8 | 16 | 32 | 64 | 128 |
|---|---|---|---|---|---|---|---|
| Avg. Acc (%) | 55.5 | 59.3 | 60.1 | 62.3 | 62.5 | 62.6 | 62.6 |

### B.10. Computational Efficiency and Real-Time Adaptation

To comprehensively assess the practicality of BETA, we analyze efficiency across two dimensions: API costs (query complexity) and local computational overhead. We further validate performance under a strict real-time streaming protocol, following (Alfarra et al.).

**Detailed Efficiency Breakdown.** We conducted a granular breakdown of wall-clock latency and resource usage using a single NVIDIA RTX 3090 GPU. As summarized in Table 5, we compare BETA against baselines including ZOO-SPSA-GC[†] and Test-Time Augmentation (TT-Aug) (Shanmugam et al., 2021).

The analysis yields two critical insights. First, **local computation is negligible** compared to API latency. While BETA introduces a local steering model (ViT-Small), it requires only 2.6GB of GPU memory—feasible for consumer-grade hardware—and adds a trivial 0.003s overhead per image for the backward pass. The primary bottleneck in black-box adaptation is the API forward pass ($T_{API} \approx 0.045s$), which is dominated by network latency. Second, **API calls dominate total latency**. Methods relying on multiple queries per image suffer from severe slowdowns. ZOO (16 calls) and TT-Aug (64 calls) are approximately $9.4\times$ $(0.450s)$ and $37.5\times$ $(1.800s)$ slower than BETA per image, respectively. This clarifies the context for "backpropagation-free" approaches in this setting: eliminating the local backward pass $(0.003s)$ provides no practical speed benefit when the total time is dictated by the mandatory API call $(0.045s)$.

**Computationally Constrained Evaluation.** To further rigorously test feasibility in streaming scenarios, we adopt the *Realistic Evaluation Protocol* from (Alfarra et al.). This protocol penalizes methods that cannot keep pace with a data stream arriving at the API's maximum throughput speed ($r = 1$ img$/T_{API}$).

We define the relative adaptation cost based on the total processing time per step: $T_{Step} = \max(T_{API}, T_{Local\_Fwd}) + T_{Local\_Bwd}$. Crucially, BETA allows for the parallelization of the local steering model's forward pass with the API query latency. Since $T_{API} \gg T_{Local\_Fwd}$, the local forward cost is effectively hidden, leaving only the negligible backward pass. Consequently, BETA maintains a relative cost $\mathcal{C} \approx 1$, allowing it to adapt to virtually 100% of the data stream. In contrast, query-intensive methods like ZOO incur massive adaptation lag ($\mathcal{C} \gg 1$), forcing them to skip adaptation for the majority of samples to maintain throughput.

The results in Table 15 demonstrate the impact of this constraint. Under strict real-time conditions, ZOO's performance drops to 54.3% (worse than the Source), as it updates too infrequently. BETA, however, maintains an accuracy of 62.5%, confirming it is a viable solution for real-time black-box adaptation.

## C. Use of Large Language Models

We used a Large Language Model to assist with language polishing and improving the readability of this manuscript. The authors are fully responsible for all research ideas, experimental results, and claims presented in this paper.

*Table 15.* Evaluation under computational time constraints. "Offline" assumes unlimited time; "Online" simulates realistic streaming where slow methods skip samples.

| Method | Offline Acc (%) | Online Acc (%) |
|---|---|---|
| Source | 55.5 | 55.5 |
| LAME | 54.1 | 54.1 |
| ZOO | 56.0 | 54.3 |
| **BETA (Ours)** | **62.6** | **62.5** |

# D. Limitations

While BETA demonstrates strong performance and efficiency, its effectiveness is connected to the choice of the local steering model. In the current landscape, where most large-scale models are Transformer-based, our method is highly applicable, as finding a steering model with a similar architecture is straightforward. However, the performance could be suboptimal if the architectures of the steering and target models differ significantly. Although our experiments show that cross-architecture adaptation is effective (e.g., a CNN steering a Transformer), the improvements are slightly less pronounced than when using architecturally similar models. Another avenue for future research is extending this framework beyond classification to more versatile, generative tasks. Investigating how to adapt the harmonized objective for generative outputs, where the prediction space is vast and unstructured, would be a valuable next step.

