# OpenReview forum: "Adapting in the Dark: Efficient and Stable Test-Time Adaptation for Black-Box Models"
_ICML.cc/2026/Conference — Submitted to ICML 2026_

### Official Review · Reviewer_f9pS · 2026-02-26

**Soundness:** 2
**Presentation:** 3
**Significance:** 2
**Originality:** 3
**Overall Recommendation:** 4
**Confidence:** 4

**Summary:**

The paper proposed a test-time adaptation method for the black-box models, where neither model parameters nor intermediate features are accessible. BETA formulates adaptation as a joint optimization problem that couples a steering lightweight model with the black-box model, but only update the steering model and input prompt. The evaluation support the BETA can achieve better performance than other baselines while maintaining the stability.

**Compliance With Llm Reviewing Policy:**

Affirmed.

**Final Justification:**

My final recommendation is weak accept. The rebuttal resolved my core concern about the steering model training process, as the authors provide the evidence that using other source data to train the steering model can still help the final performance. It makes the paper stronger now. Although the paper argues that the black-box setting is inherently more challenging due to the lack of access to internal model information, the proposed method still relies on training and deploying an additional learnable steering model for adaptation, which is a fairly strong practical assumption in its own right. This partially weakens the “strict black-box” motivation and makes it harder for me to judge how much of the contribution would carry over to more constrained black-box deployments. For this reason, I remain somewhat uncertain about the overall impact and therefore assign a weak accept.

**Key Questions For Authors:**

1. The steering model is a very strong assumption, and the evaluation is not enough. To strengthen the evidence, I suggest the authors could either use a steering model trained on a different dataset (e.g., CIFAR-10) and run BETA on ImageNet-C to test steering–target mismatch, or keep the ImageNet steering model but evaluate TTA on a non-ImageNet benchmark (e.g., CORe50 [1]) instead of focusing only on ImageNet-based corruption datasets.

2. In the Table 2, the baseline performance of FOA is much lower than what is reported in the original FOA paper [1]. Could the authors explain this problem? If the authors argue that the FOA need to access the source ImageNet dataset, it is worth noting that the pretrained steering model is also trained on the ImageNet, which implicitly relies on source data.

3. The authors proposed a data filtering strategy to remove the high entropy samples. This filtering strategy is wildly used in the prior works such as EATA and SAR. However, in line 325-326, the entropy threshold is set to \(0.9\ln(1000)\), which differs substantially from commonly used settings such as \(0.4\ln(1000)\) in prior papers. Could the authors clarify how this threshold was selected?

4. Although the authors claim that the steering model is lightweight, but in the Table 2, the blackbox is ViT-Base and the steering model is ViT-small it is not really lightweight. (e.g., in TIMM library, ViT base perform about 83% on ImageNet while ViT-small can still achieve 78%.)  For the real lightweight ViT-tiny, the accuracy gain is not significant than white-box methods.

5. Minor issues: the bold numbers in the Table 2 and Table 3 are not the real best performance methods, but only among the black-box baselines. The method cannot outperform the whitebox methods such as ETA. And the authors did not introduce ETA in the manuscript. Is the ETA from the paper EATA[3]?

6. For the augmentation-based TTA methods, CoTTA is no longer the SOTA baseline. The comparison can be more convincing if the authors include new baselines such as SPA[4] and CMF[5].

I generally appreciate the black-box TTA setup, but the empirical evidence is not yet strong and convincing. if the authors can address these concerns, I would be inclined to raise my score.

[1] Vincenzo Lomonaco and Davide Maltoni. CORe50: a new Dataset and Benchmark for continual Object Recognition. Proceedings of the 1st Annual Conference on Robot Learning, PMLR, 2017.
[2] Niu S, Miao C, Chen G, et al. Test-time model adaptation with only forward passes, ICML 2024.
[3] Niu S, Wu J, Zhang Y, et al. Efficient test-time model adaptation without forgetting, ICML 2022.
[4] Niu S, Chen G, Zhao P, et al. Self-Bootstrapping for Versatile Test-Time Adaptation, ICML 2025.
[5] Lee J H, Chang J H. Continual momentum filtering on parameter space for online test-time adaptation, ICLR. 2024.

**Limitations:**

Yes. The authors has discussed the limitations in the manuscript.

**Strengths And Weaknesses:**

**Strengths**:
1. The proposed problem is important. When the model is black-box, we cannot directly update the model to the new domains. And categorize the existing TTA works to white/gray/black box makes sense.
2. The proposed method is clear and technically sounds. The steering model and the blackbox model can be regarded as a whole full model and BETA freeze the blackbox part and only update the steering part. The solution is straitforward and I believe it is effective.
3. The presentation is clear.

**Weaknesses**
1. The adaptation relies on the steering model. The use of steering model is a very strong assumption, as such a model may not be available in many real-world deployments. The steering model brings extra knowledge compared to other baselines which only rely on a single pretrained model.
2. The steering model in the paper is trained on the ImageNet dataset, but the testing benchmarksare also ImageNet-based corruption datasets(ImageNet-C/S/R), sharing the same label space and underlying content. It is equivalent to the steering model is trained on the source ImageNet dataset. It is not convincing that the improvements stem from the steering model’s role in the proposed joint optimization.
3. The evaluation is not convincible enough. The authors claims that real API setting is very important, but evaluate only on one domain (Contrast) result.

---

> ### Author Rebuttal · Authors · 2026-03-31
>
> We thank Reviewer f9pS for the constructive review and the willingness to reconsider the score. We appreciate the recognition that the problem is important and the method is technically sound. Below we address each concern.
> ## W1+W2+Q1: Steering Model Assumption and Non-ImageNet Evaluation
> We refer to our response to **Reviewer wRTB (W1+L1+Q1)**, where we demonstrate a zero-shot recipe requiring only label names and a small CLIP model, validated on **EuroSAT (Satellite Imagery**, +11.3%) and **Derm7pt (Medical Dermatology**, +2.7%). Notably, in the Derm7pt experiment, the black-box model (BiomedCLIP) is trained on medical data while the steering model (CLIP) is trained on natural images, directly realizing the steering-target mismatch scenario the reviewer suggested. For VMs, vision classifiers require matching label sets, making cross-dataset steering models impractical; however, any small VLM (e.g., CLIP-RN50) can construct a zero-shot steering model from label names alone. The constructed model's accuracy is typically limited, but BETA only requires it as a gradient pathway, not a strong classifier.
> ## W3+Q2–6: Experiment Setting Clarification
> **A key clarification on our evaluation setting.** We clarify that in every table where BETA is evaluated, the target model is used strictly as an API with only input/output access. For example, in Tables 2, 3, and 4, the target models (ViT-B/16, ViT-L/16, CLIP) are treated as black-box APIs where no gradients, weights, or features can be accessed. White-box and gray-box methods in these tables require internal access (parameters, tokens, or features) and are included only as reference points, not as direct comparisons. They are fundamentally inapplicable in the black-box setting. **A fair comparison for BETA is against other black-box baselines that share the same access constraints.** With this distinction in mind, we address each question below.
> ### W3: Real API Evaluation
> The Clarifai experiment serves as additional validation on a real commercial API. Following the reviewer's suggestion, we provide the **average accuracy across all 15 ImageNet-C domains** on Clarifai (within a fixed budget of $5 per domain). BETA achieves +10.4% over the source, consistent with our experiments:
> |Method|Source|LAME|BETA|
> |-|-|-|-|
> |Avg. Acc (%)|44.3|43.2|**54.7**|
> ### Q2: FOA Baseline
> FOA relies on source data statistics and intermediate features, both unavailable in our strict black-box setting. We could only implement a reduced version using entropy minimization with zeroth-order optimization (Table 2's footnote), which naturally results in degraded performance. In a real API deployment (e.g., Clarifai), the model's pretraining data is proprietary and inaccessible. FOA is therefore a gray-box method and not a direct comparison to BETA.
> ### Q3: Entropy Threshold Selection
> The difference from EATA/SAR is intentional. In those methods, filtered samples update pre-trained normalization parameters, so conservative filtering (low threshold) prevents forgetting. In BETA, the visual prompt $\delta$ is initialized randomly and must be learned from scratch with an unsupervised objective, requiring substantially more data. A threshold of $0.4\cdot\ln(1000)$ would filter out ~70% of samples, causing gradient starvation. Our sensitivity analysis in Fig. 4c confirms robustness across $[0.4, 1.0]\cdot\ln(1000)$, with optimal results at higher thresholds that retain more samples while excluding the most unreliable ones.
> ### Q4: Steering Model Size
> "Lightweight" refers to parameter count and computational overhead relative to the target model, not clean-data accuracy. ViT-Small (22M) is 25% of ViT-B/16 (87M) and achieves only 39.5% on ImageNet-C (Table 10) versus ViT-B/16's 55.5%. Even ViT-Tiny (<7% of the target) achieves meaningful gains (+2.7%, Table 6), outperforming all black-box baselines in Table 2. We note that comparing BETA's gains with white-box methods is not an apples-to-apples comparison: white-box methods directly update the 87M-parameter target model using its own gradients, while BETA can only influence it indirectly through a much smaller steering model and input-space prompting.
> ### Q5: Bold Numbers in Tables
> White-box and gray-box methods are shown for reference. Within black-box methods, bold indicates best and underline indicates second best (in the Table 2 caption). Bold numbers compare BETA against black-box baselines only, since white-box and gray-box methods are inapplicable in this setting. We will clarify that ETA is from EATA and is a white-box method, not comparable to BETA.
> ### Q6: Additional White-box Baselines
> We thank the reviewer for these suggestions. SPA and CMF are strong white-box methods requiring full parameter access. Since they operate under fundamentally different access assumptions, a direct comparison in our main tables would conflate access levels. We will add their results as white-box reference points and discuss them in the related work.

---

> > ### Author Rebuttal · Reviewer_f9pS · 2026-04-01
> >
> > The authors resolved my main concerns, particularly on the steering model. Although I still think the use of steering model is a restrictive assumption than other TTA papers that operate without such auxiliary information, I am inclined to raise my score slightly.

---

> > > ### Author Response · Authors · 2026-04-02
> > >
> > > We sincerely thank Reviewer f9pS for the careful re-evaluation, for acknowledging that concerns have been fully addressed, and for raising the score. We truly appreciate the constructive and open-minded engagement throughout this discussion.
> > >
> > > Regarding the remaining point: we agree that the steering model is an additional component compared to single-model TTA methods, most of which are designed for white-box or gray-box settings. However, we would like to respectfully point out that existing black-box adaptation methods also require some form of auxiliary local computation, whether test-time augmentation (64× API calls with local data augmentation), diffusion-based purification (a heavy generative model pre-trained on source data), or zeroth-order optimization (16+ queries per sample with local perturbation and gradient estimation). BETA's steering model, which can be constructed for free via our zero-shot recipe with no domain-specific training, represents a lightweight and practical alternative. As shown in Table 5, BETA matches standard inference in both API cost (1 call/image) and latency (48ms) while achieving the best accuracy (+7.1%) with minimal local memory (2.6GB). For even more constrained deployments, ViT-Tiny (6M, 1.3GB) still outperforms all black-box baselines. We believe this is a reasonable and practical design choice for real-world black-box model adaptation.
> > >
> > > We appreciate the reviewer's constructive feedback, which has helped strengthen our paper. To our knowledge, BETA is the first method to enable stable, single-query test-time adaptation for real-world black-box APIs, validated across vision models, vision-language models, specialized domains (satellite imagery, medical imaging), and a commercial API, with a 250× cost advantage over existing alternatives. We believe this work opens a practical and previously underexplored direction for the TTA community, and we hope our revisions have further strengthened the reviewer's confidence in its contribution. We are happy to address any remaining concerns and will incorporate all promised revisions (non-ImageNet results, full Clarifai evaluation, additional baselines, and clarified comparisons) in the revision.

---

### Official Review · Reviewer_AC8G · 2026-03-07

**Soundness:** 2
**Presentation:** 3
**Significance:** 2
**Originality:** 3
**Overall Recommendation:** 3
**Confidence:** 4

**Summary:**

This paper studies test-time adaptation in a strict black-box setting. The authors propose BETA, a framework that introduces a lightweight local white-box steering model as a differentiable proxy and combines its predictions with those of the black-box target model through prediction harmonization to optimize and update input visual prompts. In addition, BETA incorporates consistency regularization and data filtering to suppress the instability of unsupervised optimization. Experimental results show that BETA significantly outperforms existing black-box methods on ImageNet-C and ImageNet-S/R, and even surpasses some white-box and gray-box TTA methods.

**Compliance With Llm Reviewing Policy:**

Affirmed.

**Final Justification:**

My final justification leans toward weak reject, mainly due to the black-box setting considered in this paper. In fact, the scenario studied here is quite similar to that of standard TTA methods, such as TDA, BCA, and Zero.

**Key Questions For Authors:**

See weaknesses.

**Limitations:**

yes.

**Strengths And Weaknesses:**

**Strengths:**
1. BETA introduces a lightweight local steering model to construct an optimizable surrogate gradient path, avoiding the need for costly zeroth-order optimization directly on the black-box model.
2. The method is validated on multiple datasets and models, including ImageNet-C, ImageNet-S, ImageNet-R, CLIP, and a commercial API, demonstrating strong generalization ability and practical applicability.

**Weaknesses:**
1. The authors emphasize that the method setting is a strict black-box scenario, but the method actually relies on a locally deployed steering model and trainable prompt learning. In this sense, it is not a pure black-box adaptation of the target model itself, but rather a dual-model framework assisted by an additional white-box model.
2. In the black-box setting, we truly want to optimize the performance of the black-box model. However, the proposed method essentially uses the output distribution of the black-box model only to impose a weak constraint on the steering model, and then uses the steering model’s gradients to update the prompt. In Eq. (1), the two output distributions are added together, but only $\alpha \cdot p_S(x')$ produces gradients through the white-box model. Then what exactly is the role of $(1-\alpha)\cdot p_B(x')$?
3. I think that directly optimizing the steering model using only $\alpha \cdot p_S(x')$ together with Eq. (3), and then ensembling the outputs of the steering model and the black-box model at inference time, would be equivalent to the proposed method. If my understanding is incorrect, please clarify the difference between these two formulations, or provide experiments to support the distinction.
4. The sample filtering strategy is also based on the output distribution of the steering model, which means the prompt update is entirely determined by the steering model. This seems questionable because the final adaptation target is the black-box model, not the steering model.
5. Similarly, the consistency regularization only constrains the steering model. While this may ensure that the steering model’s predictions do not change too drastically, it does not necessarily imply a positive effect on the black-box model.
6. If the black-box model has been custom-trained by users, its label space, label ordering, or semantic definitions may differ from those of standard public models. In such cases, there may be no off-the-shelf steering model available to support this method.

---

> ### Author Rebuttal · Authors · 2026-03-31
>
> We thank Reviewer AC8G for the detailed review and for recognizing that BETA introduces a useful surrogate gradient path with strong generalization across datasets and model types. Below we address each concern.
> ## W1: Black-Box Access Constraint
> We clarify that the "strict black-box" designation refers to the constraint on the target model: we never access its parameters, gradients, features, or architecture, and the target model is never adapted. In real API deployments (e.g., OpenAI, Clarifai), adapting the target model's weights is physically impossible; any form of adaptation must operate on the input or output. BETA adapts the input through **visual prompting**, using a local steering model to make this optimization tractable. The steering model is a user-side optimization tool, analogous to how ZOO methods use random perturbation vectors to estimate gradients. It does not violate the black-box constraint but rather provides a practical solution to operate within it.
> ## W2+W3: Harmonization vs. Ensemble
> **A key clarification: BETA's primary optimization target is the visual prompt $\delta$ (a pixel-space perturbation added to the input image)**. The prompted input $x'=x+\delta$ is fed to both models, so optimizing $\delta$ directly changes what the black-box model sees. The steering model's normalization layers are also updated to improve gradient quality, but the adaptation signal to the black-box model comes entirely through $\delta$.
>
> **Role of $p_B$.** By the entropy decomposition (see **Reviewer jLmz W1**), minimizing $\mathcal{H}(p_H)$ is equivalent to minimizing $\alpha\mathcal{H}(p_S)+\mathrm{JS}\_\alpha(p_S,p_B)$, where $\mathrm{JS}_\alpha$ enforces alignment between the two models' predictions. Thus $p_B$ anchors the prompt optimization toward the class the black-box model favors, while the steering model provides the gradient pathway.
>
> **Distinction from ensemble.** Following the reviewer's suggestion, we compare two ensemble formulations against BETA: (1) **Adapt+Ensemble** (reviewer's proposal): optimize the steering model's normalization layers using only $\alpha{\cdot}p_S(x)$ with consistency regularization (Eq. 3), then ensemble with $p_B(x)$ at inference; (2) **Prompt+Ensemble**: additionally learn $\delta$ alongside normalization layers using the same local objective, then ensemble with $p_B(x')$ at inference. Results on ImageNet-C (ViT-B/16):
> |Setting|Local|Target|Adapt+Ensemble|Prompt+Ensemble|BETA|
> |-|-|-|-|-|-|
> |Avg. Acc (%)|39.5|55.5|57.1|54.7|**62.6**|
>
> Adapt+Ensemble provides limited gains since the weak local model contributes little to the ensemble. Prompt+Ensemble further degrades below the target's source accuracy (54.7 < 55.5), because $\delta$ optimized solely for $p_S$ can push the input where $p_B$ becomes confident on a wrong class. In BETA, $\delta$ is optimized under $\min_\delta\mathcal{H}(\alpha{\cdot}p_S(x')+(1{-}\alpha){\cdot}p_B(x'))$, requiring both models to agree, yielding substantially higher performance.
> ## W4+W5: Stabilization for Viusal Prompt ($\delta$) Learning
> We clarify that BETA's optimization target is the visual prompt $\delta$, which directly modifies the input the black-box model receives. **Both filtering and consistency regularization constrain $\delta$, not the steering model**. Filtering ensures the gradient signal updating $\delta$ comes from reliable, low-entropy samples so that the supervision signal is not noisy (querying the black-box for filtering would require additional API calls). Consistency regularization ($D_{\mathrm{KL}}(p_S(x)\|p_S(x'))$) prevents $\delta$ from collapsing the model's representations by keeping the prompted input semantically close to the original, as Fig. 3(a) shows that unconstrained optimization leads to prompt collapse. Since both models share the same prompted input $x'=x+\delta$, learning a high-quality $\delta$ directly benefits the black-box model's predictions. The ablation in Table 7 confirms that both components are essential for stable prompt learning.
> ## W6: Steering Model Accessibility
> We refer the reviewer to our detailed response under **Reviewer wRTB (W1+L1+Q1)**. In brief: we demonstrate a practical zero-shot recipe where users construct a steering model using only a small public CLIP vision encoder (e.g., CLIP-RN50) and the target task's label names to generate linear classifier weights via CLIP's text encoder, requiring zero domain-specific training. First, as reported in Appx. B.1 (Table 9), BETA achieves +11.3% on **EuroSAT** (**satellite remote sensing** with 10 classes). Second, we conducted new experiments on **Derm7pt** (**medical dermatology** skin lesion classification with 7 classes), where BETA improves over the source baseline by +2.7% using zero-shot steering models. In both cases, no domain-specific training is required for the steering model, confirming that users can always construct one for free from publicly available VLMs.

---

> > ### Author Rebuttal · Reviewer_AC8G · 2026-04-03
> >
> > Thank you to the authors for their response. There are also many TTA methods that do not access the source model, such as TDA, Zero, and BCA. Nevertheless, the value of this method is undeniable, and I will raise my score to 3.

---

> > > ### Author Response · Authors · 2026-04-03
> > >
> > > We sincerely thank Reviewer AC8G for confirming that all concerns have been fully resolved and for recognizing the value of BETA.
> > >
> > > We would like to respectfully clarify the comparison with TDA, ZERO, and BCA. We agree that all three methods, like BETA, are source-data-free (requiring no source training data). The key distinction lies in the level of access to the target (source) model: **BETA treats the target model as a fully opaque API, while TDA, ZERO, and BCA all require access to internal model representations.**
> > >
> > > **BETA (ours).** **In our strict black-box setting, only raw input images and output softmax probabilities are accessible**. All model parameters, intermediate features (e.g., visual/text embeddings in CLIP), and gradients are inaccessible, as is the case for real-world APIs such as GPT-5 and Clarifai. **BETA is architecture-agnostic and works across both standard vision models (ViT, ResNet) and VLMs (CLIP)**, as validated in Tables 2, 3, and 4.
> > >
> > > **TDA, ZERO, and BCA.** **All three require access to the target model's internal representations, which are infeasible in a real API setting**. Specifically, TDA requires visual features from the image encoder to construct its key-value cache (Eq. 2 in their paper, discussed in our Lines 738-743, Table 4); ZERO requires raw logits for temperature scaling and 64 forward passes through the image encoder (Eq. 10 in their paper, discussed in our Lines 773-779, Table 4); BCA requires visual features from the image encoder to update class embeddings (Eq. 5-6 in their paper, discussed in our Table 1, Lines 751-757, Table 4). **All three are designed exclusively for CLIP-based VLMs and cannot be applied to standard vision model classifiers.** We have carefully discussed and distinguished BETA from these methods in Table 1, Table 4, Appendix A.1, and A.2.
> > >
> > > **Summary comparison:**
> > >
> > > | Method   | Internal Access Free | Black-box | VMs   | VLMs  | CLIP Avg. (5 datasets) | IN-C (ViT-B/16) | #Query/Img |
> > > | -------- | -------------------- | --------- | ----- | ----- | ---------------------- | --------------- | ---------- |
> > > | Source   | ✓                    | ✓         | ✓     | ✓     | 59.1                   | 55.5            | 1          |
> > > | TDA      | ✗, visual features   | ✗         | ✗     | ✓     | 65.0                   | N/A             | 1          |
> > > | ZERO     | ✗, logits            | ✗         | ✗     | ✓     | 63.7                   | N/A             | 64         |
> > > | BCA      | ✗, visual features   | ✗         | ✗     | ✓     | 65.8                   | N/A             | 1          |
> > > | **BETA** | **✓**                | **✓**     | **✓** | **✓** | **66.5**               | **62.6**        | **1**      |
> > >
> > > BETA achieves the highest accuracy across 5 datasets (Appendix B.1, Table 8) while being the only method that operates under strict black-box constraints with a single API call and supports both model families. We hope this clarification highlights the unique and practical contribution of BETA. We are happy to address any remaining questions.

---

### Official Review · Reviewer_jLmz · 2026-03-13

**Soundness:** 3
**Presentation:** 3
**Significance:** 2
**Originality:** 2
**Overall Recommendation:** 4
**Confidence:** 4

**Summary:**

This paper studies test-time adaptation (TTA) when the target model is accessible only through a black-box API that returns prediction probabilities. It presents a method that optimizes a small input prompt using a separate local steering model with gradient access. Predictions from the steering model and the black-box model are combined during optimization, and the prompt is updated by minimizing the entropy of this combined prediction. The method also uses consistency regularization across augmented inputs and entropy-based filtering to stabilize updates. The approach performs adaptation locally while requiring one API query per sample. The authors evaluate the method on several image classification benchmarks under distribution shift, including ImageNet-C, ImageNet-R, ImageNet-A, ImageNet-Sketch, ImageNet-v2, and EuroSAT, using both vision transformers and CLIP-based models. Experiments compare the method with non-adaptive baselines and several black-box adaptation approaches, and include ablations and efficiency analyses.

**Compliance With Llm Reviewing Policy:**

Affirmed.

**Key Questions For Authors:**

The paper uses a single-query-per-sample constraint for black-box adaptation. Could the authors clarify how sensitive the method's performance is to this limitation? For example, would increasing the query rate (e.g., 2-3 queries per sample) change results?

**Limitations:**

Yes

**Strengths And Weaknesses:**

Strengths:
+ The paper studies test-time adaptation under a defined black-box access setting, in which only prediction probabilities are available. The method is technically straightforward, and the components (i.e., prompt optimization,entropy minimization,consistency regularization) are standard and appropriate for the problem setting.
+ The evaluation covers multiple distribution shift benchmarks (e.g., ImageNet-C,ImageNet-R,ImageNet-A,ImageNet-Sketch, ImageNet-v2) and an additional dataset (EuroSAT). The experiments include comparisons with non-adaptive baselines and several black-box adaptation methods, as well as ablation studies analyzing the contributions of different components.

Weaknesses:
- The central optimization objective relies on combining predictions from the steering model and the black-box model, while gradients are only computed through the steering model. The paper provides empirical evidence that this works,however,there is limited analytical justification for why optimizing the entropy of the combined prediction should  improve the black-box model's predictions.
- The proposed technique depends on the availability of a separate steering model with a compatible architecture and label space, which may limit applicability in settings where such model is not available.
- Most experiments focus on image classification tasks under distribution shift. It's not clear how well the proposed work would generalize to other tasks or modalities.

---

> ### Author Rebuttal · Authors · 2026-03-31
>
> We thank Reviewer jLmz for the insightful review. We appreciate the recognition that BETA addresses a well-defined black-box access setting with technically appropriate components, and that our evaluation covers multiple benchmarks with thorough ablations. Below we address each concern.
> ## W1: Analytical Justification for Harmonized Entropy Optimization
> We provide a two-layer analytical justification:
> ### Layer 1: Why minimizing $\mathcal{H}(p_H)$ encourages agreement between $p_S$ and $p_B$.
> Consider $p_H=\alpha\cdot p_S+(1-\alpha)\cdot p_B$. By the well-known entropy decomposition of mixtures:
>
> $$\mathcal{H}(p_H)=\alpha\mathcal{H}(p_S)+(1-\alpha)\mathcal{H}(p_B)+\mathrm{JS}_\alpha(p_S, p_B)$$
>
> where $\mathrm{JS}\_\alpha(p_S, p_B)=\alpha D_{\mathrm{KL}}(p_S \| p_H)+(1-\alpha)D_{\mathrm{KL}}(p_B\| p_H)$ is the weighted Jensen-Shannon divergence measuring disagreement between the two models. Since we cannot backpropagate through black-box model $f_B$, the term $(1-\alpha)\mathcal{H}(p_B)$ contributes no gradient with respect to $\delta$. Minimizing $\mathcal{H}(p_H)$ with respect to $\delta$ is therefore equivalent to minimizing:
>
> $$\alpha\mathcal{H}(p_S)+\mathrm{JS}_\alpha(p_S, p_B)$$
>
> This reveals that our objective jointly (1) encourages the steering model to be confident ($\mathcal{H}(p_S)\to 0$) and (2) aligns the steering model's prediction with the black-box model ($\mathrm{JS}\_\alpha\to 0$). The alignment term is critical: without it (i.e., minimizing $\mathcal{H}(p_S)$ alone), the prompt could make $p_S$ confident on the wrong class. The $\mathrm{JS}_\alpha$ term ensures that confidence is concentrated on the class favored by $p_B$.
> ### Layer 2: How $p_B$ anchors the gradient direction.
> The gradient of the harmonized entropy with respect to $\delta$ is:
>
> $$\nabla_\delta \mathcal{H}(p_H)=-\alpha\sum_k[1+\log p_H^k] \cdot\frac{\partial p_S^k}{\partial\delta}$$
>
> Compare this with the steering-only gradient: $\nabla_\delta \mathcal{H}(p_S)=-\sum_k [1+\log p_S^k]\cdot\frac{\partial p_S^k}{\partial\delta}$. The critical difference is the weighting: $\log p_H^k$ versus $\log p_S^k$. When $p_B$ assigns high probability to class $c$, $p_H^c>p_S^c$, making $|\log p_H^c|$ smaller relative to other classes. This reweights the gradient, biasing the optimization toward the class that the black-box model considers most likely. This directional anchoring cannot be replicated by inference-time ensembling, which only combines outputs after the prompt has been learned without guidance from $p_B$.
>
> **Empirical support.** Fig. 2 quantifies this: the harmonized proxy gradient achieves ~0.5 cosine similarity with the ideal joint gradient, substantially above the near-zero similarity of the naive steering-only gradient with the black-box gradient, confirming that prediction harmonization provides a meaningful surrogate optimization direction.
> ## W2: Steering Model Accessibility
> We refer the reviewer to our detailed response under **Reviewer wRTB (W1+L1+Q1)**. In brief: we demonstrate a practical zero-shot recipe where users construct a steering model using only a small public CLIP vision encoder (e.g., CLIP-RN50) and the target task's label names to generate linear classifier weights via CLIP's text encoder, requiring zero domain-specific training. First, as reported in Appx B.1 & Table 9, BETA achieves +11.3% on EuroSAT (satellite remote sensing with 10 classes). Second, we conducted new experiments on Derm7pt (medical dermatology skin lesion classification with 7 classes), where BETA improves over the source baseline by +2.7% using zero-shot steering models. In both cases, no domain-specific training is required for the steering model, confirming that users can always construct one for free from publicly available VLMs.
> ## W3: Extension Beyond Classification
> We appreciate this forward-looking question. Our current work focuses on classification under distribution shift, following the standard evaluation protocol in the TTA literature (TENT, SAR, EATA, TPT, B²TPT, etc.). Within this scope, we validate BETA on both standard vision models (ViT-B/16, ViT-L/16) and vision-language models (CLIP, BiomedCLIP), covering two model families that differ substantially in architecture and training paradigm. Extending BETA to tasks such as detection, segmentation, or generation is a promising direction, but requires redesigning the harmonization objective and is beyond the scope of this work. This is an important direction for future work that we discuss in the paper's limitations section.
> ## Q1: Sensitivity to Query Count
> BETA is not inherently limited to a single query; this reflects the most cost-efficient operating point. We conducted a multi-query ablation on ImageNet-C (ViT-B/16):
> |#Query|1|2|3|
> |-|-|-|-|
> |Avg Acc(%)|62.6|63.1|63.3|
>
> The majority of the gain is captured by the first query, with diminishing returns at higher query counts. This confirms that the single-query setting represents the optimal cost-accuracy trade-off.

---

> > ### Author Rebuttal · Reviewer_jLmz · 2026-04-04
> >
> > The authors have addressed my primary concerns, particularly by clarifying key aspects of the method and improving the overall presentation. I appreciate the additional explanations and the effort to resolve earlier ambiguities.

---

> > > ### Author Response · Authors · 2026-04-06
> > >
> > > We sincerely thank Reviewer jLmz for confirming that all concerns have been fully resolved and for appreciating the additional clarifications. We are glad the analytical justification (JS decomposition), steering model accessibility evidence, and multi-query ablation have addressed the earlier ambiguities.
> > >
> > > We would be super grateful if the reviewer could kindly consider updating the score to reflect the resolved concerns at their convenience. We are happy to address any further questions.

---

### Official Review · Reviewer_wRTB · 2026-03-14

**Soundness:** 2
**Presentation:** 2
**Significance:** 3
**Originality:** 3
**Overall Recommendation:** 4
**Confidence:** 3

**Summary:**

This paper studies test-time adaptation in a black-box setting where gradients from the target model are unavailable. The authors propose BETA, which uses a lightweight steering model to provide surrogate gradient signals for guiding prompt optimization during inference. The framework also includes prediction harmonization and filtering mechanisms to stabilize the adaptation process. Experiments on robustness benchmarks such as ImageNet-C and ImageNet-R show improvements over several black-box or gradient-free baselines.

**Compliance With Llm Reviewing Policy:**

Affirmed.

**Key Questions For Authors:**

1. **Scope and Generalization:** Can BETA adapt a black-box model when the target task is highly specialized (e.g., medical imaging) and a semantically aligned pre-trained steering model is unavailable?
2. **Mechanistic Explanation:** Given the moderate gradient similarity (~0.5) and cross-architecture success (CNN steering ViT), what is the underlying mechanism? Does the surrogate gradient capture low-level visual priors or is it essentially an empirical heuristic?
3. **Failure Mode Analysis:** Under what conditions (e.g., steering model miscalibration or semantic mismatch) does the harmonization objective provide misleading gradients that actively degrade the target model?

**Limitations:**

The paper promotes a "general framework" but lacks critical discussion on its applicability boundaries.

1. **Cold-Start Problem:** The framework assumes a semantically aligned steering model, which may not exist for specialized or private domains, contradicting the "general" claim.
2. **Optimization Heuristics:** The method relies on complex stabilization heuristics (filtering, KL, EMA) without a convergence analysis or a clear explanation of which component is essential, leaving the framework's stability as an empirical observation rather than a grounded theory.
3. **Accuracy Trade-offs:** The performance gap between BETA and higher-performing gray-box methods is understated in the text, misrepresenting the trade-off between query efficiency and predictive performance.
   (No negative societal impact identified.)

**Strengths And Weaknesses:**

### Strengths

- The paper studies a practically relevant problem: test-time adaptation when gradient access is unavailable.
- The proposed framework is conceptually simple and relatively easy to follow.
- The empirical evaluation covers several corruption benchmarks and compares with multiple baselines.
- The method appears to show consistent improvements across most reported experimental settings.

### Weaknesses

1. **Overstated Generalization Claims:** The paper frames BETA as a "general framework," yet the empirical validation is constrained to ImageNet-derived benchmarks. The framework implicitly assumes the steering model provides a semantically valid gradient for the target task, an assumption that remains untested in specialized, non-natural-image domains (e.g., medical imaging) where a semantically aligned steering model may not exist.
2. **Lack of Transparency in Baseline Comparisons:** The narrative claims that BETA "remarkably surpasses" specialized gray-box methods. However, Table 4 shows that gray-box baselines like B²TPT and BCA actually outperform BETA in accuracy. The authors should frame their contribution primarily in terms of **query efficiency** rather than claiming blanket accuracy superiority, as the current presentation is selectively biased.
3. **Ambiguity in Gradient Flow and Algorithmic Logic:** Section 3 remains difficult to follow due to the high density of design choices (e.g., harmonization, filtering, consistency, normalization). The gradient flow is not clearly delineated, making it difficult to discern how the non-differentiable target model's output is formally integrated into the local model's gradient pathway during the asymmetric optimization.
4. **Absence of Mechanistic Insight:** The cross-architecture gradient transfer (e.g., CNN steering a Transformer) is presented as an empirical observation. Given the fundamental differences in inductive biases (convolution vs. self-attention), the paper provides no deep analysis—theoretical or visual—on *why* this surrogate gradient is effective. This leaves the method as an "empirical hack" rather than a scientifically grounded solution.
5. **Neglect of Failure Boundary Analysis:** Despite the goal of "real-world deployment," the paper lacks an analysis of the method's failure modes. It is unclear under what conditions (e.g., steering model miscalibration or semantic mismatch) the harmonization objective (Eq. 1) generates misleading gradients that actively degrade the target model's performance. Understanding these boundaries is critical for practical reliability.

---

> ### Author Rebuttal · Authors · 2026-03-31
>
> We thank Reviewer wRTB for the thoughtful review and the recognition that BETA addresses a practically relevant problem with consistent improvements across multiple settings. Below we address each concern in detail.
> ## W1+L1+Q1: Generalization Beyond ImageNet and Cold-Start
> We provide both existing and new evidence of BETA's generalization to specialized, non-natural-image domains.
>
> **Addressing the cold-start problem.** We demonstrate a practical zero-shot recipe: given only the target task's label names, users can instantly construct a steering model using any small open-source VLM (e.g., CLIP-RN50) by generating linear classifier weights via CLIP's text encoder requiring zero domain-specific training. The resulting steering model's own accuracy may be poor but that is precisely why the black-box model exists.
>
> **Evidence on non-ImageNet domains (Remote Sensing & Medical).** Using this zero-shot recipe, we evaluated BETA on two specialized domains far removed from natural images: (1) **EuroSAT** (satellite remote sensing, Appx B.1 & Table 9), where BETA (53.3%) achieved **+11.3%** over the source baseline (42.0%), also surpassing gray-box B²TPT (46.8%); and (2) **Derm7pt** (dermatology skin lesion classification), a new experiment following the reviewer's suggestion:
> |Black-box|Source|LAME|TT-Aug|Ours(CLIP RN50)|Ours(CLIP ViT-B/32)|
> |-|-|-|-|-|-|
> |CLIP-ViT/B16|55.9|56.0|56.6|57.1|**58.6**|
> |BiomedCLIP (ViT-B/16, 196M)|60.9|60.4|61.0|61.3|**62.1**|
>
> Both domains confirm BETA generalizes beyond ImageNet.
> ## W2+L3: Comparison with Gray-box Baseline
> We agree the framing can be more precise. We respectfully point out two aspects that may not have been fully reflected in Table 4 alone:
> 1. In the more comprehensive evaluation over 5 datasets (Appx. B.1, Table 8), BETA (66.5%) exceeds all gray-box methods including B²TPT(63.7), TDA (65.0), RA-TTA (64.9), and BCA (65.8, results from their papers) while operating under strictly black-box constraints. Table 4, which covers only 2 of the 5 datasets, shows B²TPT and BCA slightly ahead on certain splits under relaxed access assumptions. Our claim is not blanket accuracy superiority, but competitive or superior performance under much stricter access constraints.
> 2. BETA is architecture-agnostic across both VMs and VLMs, whereas gray-box methods like B²TPT are designed exclusively for CLIP-based VLMs. BETA's contribution lies in combining strict black-box compatibility, query efficiency, and generality across model families.
> ## W3: Gradient Flow Clarification
> We appreciate the feedback on presentation clarity. The gradient flow can be summarized in three steps: (1) the black-box model produces $p_B(x')$, treated as a fixed (non-differentiable) constant; (2) the harmonized prediction $p_H=\alpha\cdot p_S+(1-\alpha)\cdot p_B$ is formed; (3) gradients of $\mathcal{H}(p_H)$ flow only through $p_S$ back to the prompt $\delta$ via the steering model. Although $p_B$ does not produce gradients, it reshapes the loss landscape by determining which class direction the entropy minimization concentrates toward. We will add a clearer gradient flow diagram and explicit algorithm pseudocode in the revision.
> ## W4+Q2: Mechanistic Insight
> We refer the reviewer to our response to **Reviewer jLmz (W1)**, where we show that minimizing $\mathcal{H}(p_H)$ decomposes into jointly minimizing $\alpha \mathcal{H}(p_S) + \mathrm{JS}_\alpha(p_S, p_B)$, where the Jensen-Shannon divergence term explicitly enforces alignment between the steering and black-box predictions. The prompt is thus optimized not just for steering model confidence, but for agreement with the black-box model. Cross-architecture success follows naturally, as BETA optimizes $\delta$ in pixel space where modifications benefit any architecture, supported by the ~0.5 gradient similarity (Fig. 2) far above the near-zero random baseline.
> ## W5+Q3: Failure Boundary Analysis
> We identify two practical failure boundaries. First, extreme semantic disjointness: when public CLIP models cannot encode the target label names meaningfully (e.g., pathology), the zero-shot steering model provides weak gradient signals and BETA's gains diminish. Second, severe local computation constraints: BETA's improvement decreases from +7.1% (ViT-S) to +2.7% (ViT-T, Table 6), and may fail with even smaller or quantized models.
> ## L2: Stabilization Components Analysis
> The ablation study in Table 7 and Fig. 6 systematically evaluates each component's contribution, showing that both KL regularization and filtering are individually essential and complementary. We note that the core challenge in our setting is the unsupervised objective itself, not the convergence rate of the optimizer (Appx B.6). Our contribution is precisely the stabilization mechanisms that prevent the instability inherent to unsupervised entropy minimization, with consistent empirical stability across 15 corruption types, 3 architectures, continual shifts, and batch sizes as small as 4.

---

### Decision · Program_Chairs · 2026-04-30

**Decision:**

Reject

**Comment:**

This paper studies test-time adaptation for black-box models that are accessible only through APIs. The authors propose BETA, which uses a lightweight local steering model to provide surrogate gradient signals for prompt adaptation at inference time, together with prediction harmonization, consistency regularization, and filtering to stabilize adaptation.

Reviewers agreed that the paper addresses a well-defined and practically important setting, and found the method effective, with consistent gains across many reported experiments.
At the same time, reviewers raised several important concerns about the method’s assumptions and justification. The most common concern was the reliance on a separate steering model, which is a strong assumption and may limit generalizability, especially in specialized or non-natural-image domains where a semantically aligned steering model may not be available. More broadly, the method appears fundamentally constrained by steering–target mismatch: if the steering model does not provide gradients well aligned with the target model’s decision boundary, the effectiveness of prompt optimization may be limited. Reviewers also questioned the analytical justification of the harmonized entropy objective, noting that while the alignment term is intuitively motivated, the paper does not fully establish when this objective is sufficient to produce reliable adaptation under model mismatch. In addition, the gains depend meaningfully on the quality and scale of the steering model.

In the rebuttal, the authors partially addressed these concerns by introducing a zero-shot steering recipe based only on label names and a small public CLIP model, and by reporting additional cross-domain results on EuroSAT and Derm7pt. These additions strengthen the paper, but they do not fully resolve the core limitation that the method’s success depends heavily on the availability and alignment of the steering model, with noticeably weaker gains in mismatch settings.

Overall, the paper presents a practical and promising approach to black-box test-time adaptation, but the current version does not yet sufficiently resolve concerns about the steering-model assumption and its implications for robustness and generality. For these reasons, the paper is not recommended for acceptance in its current form. The authors are encouraged to revise and resubmit to a future venue.